# Distributionally Robust Linear Quadratic Control

**Bahar Taşkesen**
EPFL
bahar.taskesen@epfl.ch

**Dan A. Iancu**
Stanford University
daniancu@stanford.edu

**Çağıl Koçyiğit**
University of Luxembourg
cagil.kocyigit@uni.lu

**Daniel Kuhn**
EPFL
daniel.kuhn@epfl.ch

## Abstract

Linear-Quadratic-Gaussian (LQG) control is a fundamental control paradigm that has been studied and applied in various fields such as engineering, computer science, economics, and neuroscience. It involves controlling a system with linear dynamics and imperfect observations, subject to additive noise, with the goal of minimizing a quadratic cost function depending on the state and control variables. In this work, we consider a generalization of the discrete-time, finite-horizon LQG problem, where the noise distributions are unknown and belong to Wasserstein ambiguity sets centered at nominal (Gaussian) distributions. The objective is to minimize a worst-case cost across all distributions in the ambiguity set, including non-Gaussian distributions. Despite the added complexity, we prove that a control policy that is linear in the observations is optimal, as in the classic LQG problem. We propose a numerical solution method that efficiently characterizes this optimal control policy. Our method uses the Frank-Wolfe algorithm to identify the least-favorable distributions within the Wasserstein ambiguity sets and computes the controller's optimal policy using Kalman filter estimation under these distributions.

## 1. Introduction

The Linear Quadratic Regulator (LQR) is a classic control problem that has served as a building block for numerous applications in engineering and computer science [3, 12], economics [29], or neuroscience [47]. It involves controlling a system with linear dynamics and imperfect observations affected by additive noise, with the goal of minimizing a quadratic state and control cost. Under the assumption that noise terms are independent and normally distributed (a case referred to as Linear-Quadratic-Gaussian, or LQG), it is well known that the optimal control policy depends linearly on the observations and can be obtained efficiently by using the Kalman filtering procedure and dynamic programming [8].

Motivated by practical settings where noise distributions may not be readily available or may not be Gaussian, this paper considers a discrete-time, finite-horizon generalization of the LQG setting where noise distributions are unknown and are chosen adversarially from ambiguity sets characterized by a Wasserstein distance and centered around nominal (Gaussian) distributions.

We show that, even under distributional ambiguity, the optimal control policy remains linear in the system's observations. Our proof is novel and does not rely on traditional recursive dynamic programming arguments. Instead, we re-parametrize the control policy in terms of the purified state observations and we derive an upper bound for the resulting minimax formulation by relaxing the ambiguity set (from a Wasserstein ball into a Gelbrich ball) while simultaneously restricting the controller to linear dependencies. We then use convex duality to prove that this upper bound matches

37th Conference on Neural Information Processing Systems (NeurIPS 2023).

a lower bound obtained by restricting the ambiguity set in the dual of the minimax formulation. This implies the optimality of linear output feedback controllers, thus generalizing the classic results to a distributionally robust setting.

We also find that the worst-case distribution is actually Gaussian, which leads to a very efficient algorithm for finding optimal controllers. Specifically, we propose an algorithm based on the Frank-Wolfe first-order method that at every step solves sub-problems corresponding to classic LQG control problems, using Kalman filtering and dynamic programming. We show that this algorithm enjoys a sublinear convergence rate and is susceptible to parallelization. Lastly, we implement the algorithm leveraging PyTorch's automatic differentiation module and we find that it yields uniformly lower runtimes than a direct method (based on solving semidefinite programs) across all problem horizons.

## 1.1. Literature Review

This paper is related to the ample literature in control theory and engineering aimed at designing controllers that are robust to noise. The classic LQR/LQG theory, developed in the 1960s, examined linear dynamical systems in either time or frequency domain, seeking to minimize a combination of quadratic state and control costs (in time-domain) or the $\mathcal{H}_2$ norm of the system's transfer function (in frequency domain). Motivated by findings that LQG controllers do not provide the guaranteed robust stability properties of LQR controllers [15], much effort has been devoted subsequently to designing controllers that are robust to worst-case perturbations, typically evaluated in terms of the $\mathcal{H}_\infty$ norm of the system's transfer function (see, e.g., [16, 53] for a comprehensive review of $\mathcal{H}_\infty$ and $\mathcal{H}_2$ controllers). Because $\mathcal{H}_\infty$ controllers tend to be overly conservative [32], various approaches have been proposed to balance the performance of nominal and robust controllers, e.g., by combining $\mathcal{H}_2$ and $\mathcal{H}_\infty$ approaches [7, 17]. A parallel stream of literature has considered risk-sensitive control [51], which minimizes an entropic risk measure instead of the expected quadratic cost. Although risk-sensitive control has a distributionally robust flavor (as the entropic risk measure is equivalent to a distributionally robust quadratic objective penalized via Kullback-Leibler divergence), risk-sensitive control models do not admit a distributionally robust formulation because the entropic risk measure is convex, but not coherent [22]. In contrast, our distributionally robust model provides a direct interpretation of the exact set of noise distributions against which the controller provides safeguards, and leads to a computationally tractable framework for finding the optimal controller.

In this sense, our work is more directly related to the literature on distributionally robust control, which seeks controllers that minimize expected costs under worst-case noise distributions [11, 33, 34, 41, 50, 52]. Closest to our work are [28, 33]. [33] proves the optimality of linear state-feedback control policies for a related minimax LQR model with a Wasserstein distance but with *perfect* state observations. With perfect observations, the optimal policies in the classic LQR formulation are independent of the noise distribution and are thus inherently already robust, so considering imperfect observations is what makes the problem significantly more challenging in our case. [28] studies a minimax formulation based on the Wasserstein distance with both state and observation noise but without any control policy, and focuses solely on the problem of estimating the states. Several papers have considered robust formulations with imperfect observations but for constrained systems [5, 6, 34], which are more challenging; the common approach is to restrict attention to linear feedback policies for computational tractability, and without proving their optimality.

Also related is the recent literature stream on distributionally robust optimization using the Wasserstein distance [36]. Within this stream, the closest work is [38, 44], which consider the problem of minimax mean-squared-error estimation when ambiguity is modeled with a Wasserstein distance from a nominal Gaussian distribution. Our proof builds on some ideas from these papers (e.g., relying on the Gelbrich distance in the construction of the upper bound), which it combines with ideas from control theory on purified output-feedback to obtain the overall construction. Also related is [2], which studies multistage distributionally robust problems with ambiguity sets given by a nested Wasserstein distance for stochastic processes and identifies computationally tractable cases. For a broader overview of developments related to optimal transport and Wasserstein distance with an emphasis on computational tractability and applications in machine learning, we refer to [42].

Finally, our paper is also related to literature that documents the optimality of linear/affine policies in (distributionally) robust dynamic optimization models. [10, 30] prove optimality for one-dimensional linear systems affected by additive noise and with perfect state observations, but with general (convex) state and/or control costs, [27, 49] provide computationally tractable approaches to quantifying the

suboptimality of affine controllers in finite or infinite-horizon settings, and [9, 21, 25] characterize the performance of affine policies in two-stage (distributionally) robust dynamic models.

*Notation.* All random objects are defined on a probability space $(\Omega, \mathcal{F}, \mathbb{P})$. Thus, the distribution of any random vector $\xi : \Omega \to \mathbb{R}^d$ is given by the pushforward distribution $\mathbb{P}_\xi = \mathbb{P} \circ \xi^{-1}$ of $\mathbb{P}$ with respect to $\xi$. The expectation under $\mathbb{P}$ is denoted by $\mathbb{E}_{\mathbb{P}}[\cdot]$. For any $t \in \mathbb{Z}_+$, we set $[t] = \{0, \ldots, t\}$.

## 2. Problem Definition

We consider a discrete-time linear dynamical system

$$x_{t+1} = A_t x_t + B_t u_t + w_t \quad \forall t \in [T-1] \tag{1}$$

with states $x_t \in \mathbb{R}^n$, control inputs $u_t \in \mathbb{R}^m$, process noise $w_t \in \mathbb{R}^n$ and system matrices $A_t \in \mathbb{R}^{n \times n}$ and $B_t \in \mathbb{R}^{n \times m}$. The controller only has access to imperfect state measurements

$$y_t = C_t x_t + v_t \quad \forall t \in [T-1] \tag{2}$$

corrupted by observation noise $v_t \in \mathbb{R}^p$, where $C_t \in \mathbb{R}^{p \times n}$ and usually $p \leq n$ (so that observing $y_t$ would not allow reconstructing $x_t$ even if there were no observation noise). The control inputs $u_t$ are causal, i.e., depend on the past observations $y_0, \ldots, y_t$ but not on the future observations $y_{t+1}, \ldots, y_{T-1}$. More precisely, the set of feasible control inputs $\mathcal{U}_y$ is the set of random vectors $(u_0, u_1, \ldots, u_{T-1})$ where for every $t$ there exists a measurable control policy $\varphi_t : \mathbb{R}^{p(t+1)} \to \mathbb{R}^m$ such that $u_t = \varphi_t(y_0, \ldots, y_t)$. Controlling the system generates costs that depend quadratically on the states and the controls:

$$J = \sum_{t=0}^{T-1} (x_t^\top Q_t x_t + u_t^\top R_t u_t) + x_T^\top Q_T x_T, \tag{3}$$

where $Q_t \in \mathbb{S}_+^n$ and $R_t \in \mathbb{S}_{++}^m$ represent the state and input cost matrices, respectively. The exogenous random vectors $x_0$, $\{w_t\}_{t=0}^{T-1}$ and $\{v_t\}_{t=0}^{T-1}$ are mutually independent and follow probability distributions given by $\mathbb{P}_{x_0}$, $\{\mathbb{P}_{w_t}\}_{t=0}^{T-1}$, and $\{\mathbb{P}_{v_t}\}_{t=0}^{T-1}$, respectively. As the control inputs are causal, the system equations (2) imply that $x_t$, $u_t$ and $y_t$ can be expressed as measurable functions of the exogenous uncertainties $x_0$ as well as $w_s$ and $v_s$, $s \in [t]$, for every $t$. From now on we may thus assume without loss of generality that $\Omega = \mathbb{R}^n \times \mathbb{R}^{n \times T} \times \mathbb{R}^{p \times T}$ is the space of realizations of the exogenous uncertainties, $\mathcal{F}$ is the Borel $\sigma$-algebra on $\Omega$ and $\mathbb{P} = \mathbb{P}_{x_0} \otimes (\otimes_{t=0}^{T-1} \mathbb{P}_{w_t}) \otimes (\otimes_{t=0}^{T} \mathbb{P}_{v_t})$, where $\mathbb{P}_1 \otimes \mathbb{P}_2$ denotes the independent coupling of the distributions $\mathbb{P}_1$ and $\mathbb{P}_2$.

In this context, the classic LQG model assumes that $\mathbb{P}$ is known and Gaussian, and seeks $u \in \mathcal{U}_y$ that minimizes $\mathbb{E}_{\mathbb{P}}[J]$. Appendix §A reviews the standard approach for computing optimal control inputs by estimating states through Kalman filtering techniques and using dynamic programming.

In contrast, we assume that $\mathbb{P}$ is only known to belong to an ambiguity set $\mathcal{W}$, and we formulate a distributionally robust LQG problem that seeks $u \in \mathcal{U}_y$ to minimize the worst-case expected cost:

$$\max_{\mathbb{P} \in \mathcal{W}} \mathbb{E}_{\mathbb{P}} \left[ \sum_{t=0}^{T-1} (x_t^\top Q_t x_t + u_t^\top R_t u_t) + x_T^\top Q_T x_T \right]. \tag{4}$$

We construct the ambiguity set $\mathcal{W}$ as a ball based on the Wasserstein distance. Specifically, we assume that a *nominal* Gaussian distribution $\hat{\mathbb{P}} = \hat{\mathbb{P}}_{x_0} \otimes (\otimes_{t=0}^{T-1} \hat{\mathbb{P}}_{w_t}) \otimes (\otimes_{t=0}^{T} \hat{\mathbb{P}}_{v_t})$ is available so that $\hat{\mathbb{P}}_{x_0} = \mathcal{N}(0, \hat{X}_0)$, $\hat{\mathbb{P}}_{w_t} = \mathcal{N}(0, \hat{W}_t)$, and $\hat{\mathbb{P}}_{v_t} = \mathcal{N}(0, \hat{V}_t)$ for all $t \in [T-1]$, and $\mathcal{W}$ is given by:

$$\mathcal{W} = \mathcal{W}_{x_0} \otimes (\otimes_{t=0}^{T-1} \mathcal{W}_{w_t}) \otimes (\otimes_{t=0}^{T-1} \mathcal{W}_{v_t}),$$

where

$$\mathcal{W}_{x_0} = \{\mathbb{P}_{x_0} \in \mathcal{P}(\mathbb{R}^n) : \mathrm{W}(\hat{\mathbb{P}}_{x_0}, \mathbb{P}_{x_0}) \leq \rho_{x_0}, \ \mathbb{E}_{\mathbb{P}_{x_0}}[x_0] = 0\}$$
$$\mathcal{W}_{w_t} = \{\mathbb{P}_{w_t} \in \mathcal{P}(\mathbb{R}^n) : \mathrm{W}(\hat{\mathbb{P}}_{w_t}, \mathbb{P}_{w_t}) \leq \rho_{w_t}, \ \mathbb{E}_{\mathbb{P}_{w_t}}[w_t] = 0\}$$
$$\mathcal{W}_{v_t} = \{\mathbb{P}_{v_t} \in \mathcal{P}(\mathbb{R}^m) : \mathrm{W}(\hat{\mathbb{P}}_{v_t}, \mathbb{P}_{v_t}) \leq \rho_{v_t}, \ \mathbb{E}_{\mathbb{P}_{v_t}}[v_t] = 0\},$$

and $\mathrm{W}$ is the 2-Wasserstein distance. Thus, by construction, all exogenous random variables $x_0, w_0, \ldots, w_{T-1}, v_0, \ldots, v_{T-1}$ are independent under every distribution in $\mathcal{W}$.

**Definition 1** (2-Wasserstein distance). *The 2-Wasserstein distance between two distributions $\mathbb{P}_1$ and $\mathbb{P}_2$ on $\mathbb{R}^d$ with finite second moments is given by*

$$W(\mathbb{P}_1, \mathbb{P}_2) = \left( \inf_{\pi \in \Pi(\mathbb{P}_1, \mathbb{P}_2)} \int_{\mathbb{R}^d \times \mathbb{R}^d} \|\xi_1 - \xi_2\|_2^2 \, \pi(d\xi_1, d\xi_2) \right)^{\frac{1}{2}},$$

*where $\Pi(\mathbb{P}_1, \mathbb{P}_2)$ denotes the set of all couplings, that is, all joint distributions of the random variables $\xi_1$ and $\xi_2$ with marginal distributions $\mathbb{P}_1$ and $\mathbb{P}_2$, respectively.*

Our model strictly generalizes the classic LQG setting,[1] which can be recovered by choosing $\rho_{x_0} = \rho_{w_t} = \rho_{v_t} = 0$. The parameters $\rho$ thus allow quantifying the uncertainty about the nominal model and building robustness to mis-specification. We emphasize that the Wasserstein ambiguity set $\mathcal{W}$ contains many non-Gaussian distributions and it is not readily obvious that the worst-case distribution in (4) is in fact Gaussian. However, the set $\mathcal{W}$ is also non-convex, as it contains only distributions under which the exogenous uncertainties are independent, which makes the distributionally robust LQG problem potentially difficult to solve.

## 3. Nash Equilibrium and Optimality of Linear Output Feedback Controllers

We henceforth view the distributionally robust LQG problem as a zero-sum game between the controller, who chooses causal control inputs, and nature, who chooses a distribution $\mathbb{P} \in \mathcal{W}$. In this section we show that this game admits a Nash equilibrium, where nature's Nash strategy is a *Gaussian* distribution $\mathbb{P}^\star \in \mathcal{W}$ and the controller's Nash strategy is a *linear* output feedback policy based on the Kalman filter evaluated under $\mathbb{P}^\star$.

**Purified Observations.** Before outlining our proof strategy, we first simplify the problem formulation by re-parametrizing the control inputs in a more convenient form (following [5, 6, 27]). Note that the control inputs in the LQG formulation are subject to cyclic dependencies, as $u_t$ depends on $y_t$, while $y_t$ depends on $x_t$ through (2), and $x_t$ depends again on $u_t$ through (1), etc. Because these dependencies make the problem hard to analyze, it is preferable to instead consider the controls as functions of a new set of so-called *purified* observations instead of the actual observations $y_t$.

Specifically, we first introduce a fictitious *noise-free* system

$$\hat{x}_{t+1} = A_t \hat{x}_t + B_t u_t \quad \forall t \in [T-1] \quad \text{and} \quad \hat{y}_t = C_t \hat{x}_t \quad \forall t \in [T-1]$$

with states $\hat{x}_t \in \mathbb{R}^n$ and outputs $\hat{y}_t \in \mathbb{R}^p$, which is initialized by $\hat{x}_0 = 0$ and controlled by the *same* inputs $u_t$ as the original system (2). We then define the purified observation at time $t$ as $\eta_t = y_t - \hat{y}_t$ and we use $\eta = (\eta_0, \ldots, \eta_{T-1})$ to denote the trajectory of *all* purified observations.

As the inputs $u_t$ are causal, the controller can compute the fictitious state $\hat{x}_t$ and output $\hat{y}_t$ from the observations $y_0, \ldots, y_t$. Thus, $\eta_t$ is representable as a function of $y_0, \ldots, y_t$. Conversely, one can show by induction that $y_t$ can also be represented as a function of $\eta_0, \ldots, \eta_t$. Moreover, any measurable function of $y_0, \ldots, y_t$ can be expressed as a measurable function of $\eta_0, \ldots, \eta_t$ and vice-versa [27, Proposition II.1]. So if we define $\mathcal{U}_\eta$ as the set of all control inputs $(u_0, u_1, \ldots, u_{T-1})$ so that $u_t = \psi_t(\eta_0, \ldots, \eta_t)$ for some measurable function $\psi_t : \mathbb{R}^{p(t+1)} \to \mathbb{R}^m$ for every $t \in [T-1]$, the above reasoning implies that $\mathcal{U}_\eta = \mathcal{U}_y$.

In view of this, we can rewrite the distributionally robust LQG problem equivalently as

$$p^\star = \begin{cases} \min\limits_{x,u,y} & \max\limits_{\mathbb{P} \in \mathcal{W}} \mathbb{E}_\mathbb{P} \left[ u^\top R u + x^\top Q x \right] \\ \text{s.t.} & u \in \mathcal{U}_y, \ x = Hu + Gw, \ y = Cx + v \end{cases}$$

$$= \begin{cases} \min\limits_{x,u} & \max\limits_{\mathbb{P} \in \mathcal{W}} \mathbb{E}_\mathbb{P} \left[ u^\top R u + x^\top Q x \right] \\ \text{s.t.} & u \in \mathcal{U}_\eta, \ x = Hu + Gw, \end{cases} \tag{5}$$

where $x = (x_0, \ldots, x_T)$, $u = (u_0, \ldots, u_{T-1})$, $y = (y_0, \ldots, y_{T-1})$, $w = (x_0, w_0, \ldots, w_{T-1})$, $v = (v_0, \ldots, v_{T-1})$, $\eta = (\eta_0, \ldots, \eta_{T-1})$, and $R$, $Q$, $H$, $G$ and $C$ are suitable block matrices

---

[1]Our assumption that noise terms are zero-mean is consistent with the standard LQG model [8]. Requiring $\mathbb{E}_{\mathbb{P}_{x_0}}[x_0] = 0$ is assumed for clarity and without loss of generality.

(see Appendix §B for their precise definitions). The latter reformulation involving the purified observations $\eta$ is useful because these are *independent* of the inputs. Indeed, by recursively combining the equations of the original and the noise-free systems, one can show that $\eta = Dw + v$ for some block triangular matrix $D$ (see Appendix §B for its construction). This shows that the purified observations depend (linearly) on the exogenous uncertainties but *not* on the control inputs. Hence, the cyclic dependencies complicating the original system are eliminated in (5).

Subsequently, we also study the dual of (5), defined as

$$d^\star = \begin{cases} \max\limits_{\mathbb{P}\in\mathcal{W}} & \min\limits_{x,u} \ \mathbb{E}_{\mathbb{P}}\left[u^\top Ru + x^\top Qx\right] \\ \text{s.t.} & u \in \mathcal{U}_\eta, \ \ x = Hu + Gw. \end{cases} \tag{6}$$

The classic minimax inequality implies that $p^\star \geq d^\star$. If we can prove that $p^\star = d^\star$, that (5) has a solution $u^\star$ and that (6) has a solution $\mathbb{P}^\star$, then $(u^\star, \mathbb{P}^\star)$ must be a Nash equilibrium of the zero-sum game at hand [43, Theorem 2]. However, because $\mathcal{U}_\eta$ is an infinite-dimensional function space and $\mathcal{W}$ is an infinite-dimensional, non-convex set of non-parametric distributions, the existence of a Nash equilibrium (in pure strategies) is not at all evident. Instead, our proof strategy will rely on constructing an upper bound for $p^\star$ and a lower bound for $d^\star$, and showing that these match.

**Upper Bound for $p^\star$.** We obtain an upper bound for $p^\star$ by suitably *enlarging* the ambiguity set $\mathcal{W}$ and *restricting* the controllers $u_t$ to linear dependencies. We enlarge $\mathcal{W}$ by ignoring all information about the distributions in $\mathcal{W}$ except for their covariance matrices, and by replacing the Wasserstein distance with the Gelbrich distance. To that end, we first define the Gelbrich distance on the space of covariance matrices.

**Definition 2** (Gelbrich distance). *The Gelbrich distance between the two covariance matrices $\Sigma_1, \Sigma_2 \in \mathbb{S}_+^d$ is given by*

$$\mathbb{G}(\Sigma_1, \Sigma_2) = \sqrt{\mathrm{Tr}\left(\Sigma_1 + \Sigma_2 - 2\left(\Sigma_2^{\frac{1}{2}}\Sigma_1\Sigma_2^{\frac{1}{2}}\right)^{\frac{1}{2}}\right)}.$$

We are interested in the Gelbrich distance because of its close connection to the 2-Wasserstein distance. Indeed, it is known that the 2-Wasserstein distance between two distributions with zero means is bounded below by the Gelbrich distance between the respective covariance matrices.

**Proposition 3.1** (Gelbrich bound [24, Theorem 2.1]). *For any two distributions $\mathbb{P}_1$ and $\mathbb{P}_2$ on $\mathbb{R}^d$ with zero means and covariance matrices $\Sigma_1, \Sigma_2 \in \mathbb{S}_+^d$, respectively, we have $\mathbb{W}(\mathbb{P}_1, \mathbb{P}_2) \geq \mathbb{G}(\Sigma_1, \Sigma_2)$.*

Recalling that $\hat{X}_0$, $\hat{W}_t$ and $\hat{V}_t$ respectively denote the covariance matrices for $x_0, w_t$ and $v_t$ under the nominal distribution $\hat{\mathbb{P}}$, we can then define the following Gelbrich ambiguity set for the exogenous uncertainties:

$$\mathcal{G} = \mathcal{G}_{x_0} \otimes (\otimes_{t=0}^{T-1}\mathcal{G}_{w_t}) \otimes (\otimes_{t=0}^{T-1}\mathcal{G}_{v_t}),$$

where

$$\mathcal{G}_{x_0} = \{\mathbb{P}_{x_0} \in \mathcal{P}(\mathbb{R}^n) : \mathbb{E}_{\mathbb{P}_{x_0}}[x_0] = 0, \ \mathbb{E}_{\mathbb{P}}[x_0 x_0^\top] = X_0, \ \mathbb{G}(X_0, \hat{X}_0) \leq \rho_{x_0}\}$$

$$\mathcal{G}_{w_t} = \{\mathbb{P}_{w_t} \in \mathcal{P}(\mathbb{R}^n) : \mathbb{E}_{\mathbb{P}_{w_t}}[w_t] = 0, \ \mathbb{E}_{\mathbb{P}}[w_t w_t^\top] = W_t, \ \mathbb{G}(W_t, \hat{W}_t) \leq \rho_{w_t}\}$$

$$\mathcal{G}_{v_t} = \{\mathbb{P}_{v_t} \in \mathcal{P}(\mathbb{R}^m) : \mathbb{E}_{\mathbb{P}_{v_t}}[v_t] = 0, \ \mathbb{E}_{\mathbb{P}}[v_t v_t^\top] = V_t, \ \mathbb{G}(V_t, \hat{V}_t) \leq \rho_{v_t}\}.$$

By construction, the random vectors $x_0$, $\{w_t\}_{t=0}^{T-1}$ and $\{v_t\}_{t=0}^{T-1}$ are thus mutually independent under any $\mathbb{P} \in \mathcal{G}$. In addition and as a direct consequence of Proposition 3.1, $\mathcal{G}$ constitutes an outer approximation for the Wasserstein ambiguity set $\mathcal{W}$, as summarized in the next result.

**Corollary 1** (Gelbrich hull). *We have $\mathcal{W} \subseteq \mathcal{G}$.*

Because $\mathcal{G}$ covers $\mathcal{W}$, we henceforth refer to it as the *Gelbrich hull* of the Wasserstein ambiguity set $\mathcal{W}$. To finalize our construction of the upper bound on $p^\star$, we focus on linear policies[2] of the form

---

[2]Technically, the policies are affine because they include a constant term, but we retain the more common terminology that focuses on the dependencies.

$u = q + U\eta = q + U(Dw + v)$, where $q = (q_0, \ldots, q_{T-1})$, and $U$ is a block lower triangular matrix

$$
U = \begin{bmatrix} U_{0,0} & & & \\ U_{1,0} & U_{1,1} & & \\ \vdots & & \ddots & \\ U_{T-1,0} & \cdots & \cdots & U_{T-1,T-1} \end{bmatrix}. \tag{7}
$$

The block lower triangularity of $U$ ensures that the corresponding controller is causal, which in turn ensures that $u \in \mathcal{U}_\eta$. In the following, we denote by $\mathcal{U}$ the set of all block lower triangular matrices of the form (7). An upper bound on problem (5) can now be obtained by *restricting* the controller's feasible set to causal controllers that are *linear* in the purified observations $\eta$ and by *relaxing* nature's feasible set to the Gelbrich hull $\mathcal{G}$ of $\mathcal{W}$. The resulting bounding problem is given by

$$
\overline{p}^\star = \begin{cases} \min\limits_{U,q,x,u} & \max\limits_{\mathbb{P} \in \mathcal{G}} \mathbb{E}_{\mathbb{P}} \left[ u^\top R u + x^\top Q x \right] \\ \text{s.t.} & U \in \mathcal{U}, \ \ u = q + U(Dw + v), \ \ x = Hu + Gw. \end{cases} \tag{8}
$$

As we obtained (8) by restricting the feasible set of the outer minimization problem and relaxing the feasible set of the inner maximization problem in (5), it is clear that $\overline{p}^\star \geq p^\star$. Recall also that problem (5) constitutes an infinite-dimensional zero-sum game, where the agents optimize over measurable policies and non-parametric distributions, respectively. In contrast, the next proposition shows that problem (8) is equivalent to a finite-dimensional zero-sum game.

**Proposition 3.2.** *Problem* (8) *is equivalent to the optimization problem*

$$
\overline{p}^\star = \begin{cases} \min\limits_{\substack{q \in \mathbb{R}^{pT} \\ U \in \mathcal{U}}} \max\limits_{\substack{W \in \mathcal{G}_W \\ V \in \mathcal{G}_V}} & \mathrm{Tr}\left( (D^\top U^\top (R + H^\top QH)UD + 2G^\top QHUD + G^\top QG)W \right) \\ & + \mathrm{Tr}\left( (U^\top (R + H^\top QH)U)V \right) + q^\top (R + H^\top QH)q, \end{cases} \tag{9}
$$

*where*

$$
\mathcal{G}_W = \left\{ W \in \mathbb{S}_+^{n(T+1)} : \begin{array}{l} W = \mathrm{diag}(X_0, W_0, \ldots, W_{T-1}), \ X_0 \in \mathbb{S}_+^n, \ W_t \in \mathbb{S}_+^n \ \forall t \in [T-1] \\ \mathbb{G}(X_0, \hat{X}_0)^2 \leq \rho_{x_0}^2, \ \ \mathbb{G}(W_t, \hat{W}_t)^2 \leq \rho_{w_t}^2 \ \forall t \in [T-1] \end{array} \right\}
$$

$$
\mathcal{G}_V = \left\{ V \in \mathbb{S}_+^{pT} : \ V = \mathrm{diag}(V_0, \ldots, V_{T-1}), \ V_t \in \mathbb{S}_+^p, \ \mathbb{G}(V_t, \hat{V}_t)^2 \leq \rho_{v_t}^2 \ \forall t \in [T-1] \right\}.
$$

We emphasize that Proposition 3.2 remains valid even if the nominal distribution $\hat{\mathbb{P}}$ fails to be normal. Note also that, while nature's feasible set in (8) is non-convex due to the independence conditions, the sets $\mathcal{G}_W$ and $\mathcal{G}_V$ are convex and even semidefinite representable thanks to the properties of the squared Gelbrich distance.[3] By dualizing the inner maximization problem, one can therefore reformulate the minimax problem (9) as a convex semidefinite program (SDP). Even though this SDP is computationally tractable in theory, it involves $\mathcal{O}(T(mp + n^2 + p^2))$ decision variables. For practically interesting problem dimensions, it thus quickly exceeds the capabilities of existing solvers.

**Lower Bound for $d^\star$.** To derive a tractable lower bound on $d^\star$, we restrict nature's feasible set to the family $\mathcal{W}_\mathcal{N}$ of all *normal* distributions in the Wasserstein ambiguity set $\mathcal{W}$. The resulting bounding problem is thus given by

$$
\underline{d}^\star = \begin{cases} \max\limits_{\mathbb{P} \in \mathcal{W}_\mathcal{N}} & \min\limits_{x,u} \ \mathbb{E}_{\mathbb{P}} \left[ u^\top R u + x^\top Q x \right] \\ & \text{s.t.} \ \ u \in \mathcal{U}_\eta, \ \ x = Hu + Gw. \end{cases} \tag{10}
$$

As we obtained (10) by restricting the feasible set of the outer maximization problem in (6), it is clear that $\underline{d}^\star \leq d^\star$. Next, we show that (10) can be recast as a finite-dimensional zero-sum game. This result critically relies on the following known fact regarding the 2-Wasserstein distance between two *normal* distributions, which coincides with the Gelbrich distance between their covariance matrices.

**Proposition 3.3** (Tightness for normal distributions [26, Proposition 7]). *For any two normal distributions $\mathbb{P}_1 = \mathcal{N}(0, \Sigma_1)$ and $\mathbb{P}_2 = \mathcal{N}(0, \Sigma_2)$ with zero means we have $\mathbb{W}(\mathbb{P}_1, \mathbb{P}_2) = \mathbb{G}(\Sigma_1, \Sigma_2)$.*

With this, we can provide a finite-dimensional reformulation, as summarized in the next result.

---

[3]Note that the ambiguity sets $\mathcal{G}_W$ and $\mathcal{G}_V$ appearing in (9) involve the squared Gelbrich distance, $\mathbb{G}(\Sigma_1, \Sigma_2)^2$. The reason is that $\mathbb{G}(\Sigma_1, \Sigma_2)^2$ is known to be jointly convex in $\Sigma_1, \Sigma_2$ and semidefinite representable [38, Proposition 2.3], unlike the Gelbrich distance $\mathbb{G}(\Sigma_1, \Sigma_2)$ itself, which is generally non-convex.

**Proposition 3.4.** *Problem* (10) *is equivalent to the optimization problem*

$$
\underline{d}^\star =
\begin{cases}
\max\limits_{\substack{W \in \mathcal{G}_W \\ V \in \mathcal{G}_V}} \ \min\limits_{\substack{q \in \mathbb{R}^{pT} \\ U \in \mathcal{U}}} \ \mathrm{Tr}\left( (D^\top U^\top (R + H^\top QH)UD + 2G^\top QHUD + G^\top QG)W \right) \\
\qquad\qquad\qquad + \mathrm{Tr}\left( (U^\top (R + H^\top QH)U)V \right) + q^\top (R + H^\top QH)q,
\end{cases}
\tag{11}
$$

*where* $\mathcal{G}_W$ *and* $\mathcal{G}_V$ *are defined exactly as in Proposition* 3.2.

Proposition 3.4 relies on Proposition 3.3 and thus fails to hold unless $\hat{\mathbb{P}}$ is normal. Also, one can again reformulate (11) as a tractable SDP by dualizing the inner minimization problem.

**Conclusions.** Propositions 3.2 and 3.4 reveal that problems (9) and (11) are dual to each other, that is, they can be transformed into one another by interchanging minimization and maximization. The following main theorem shows that strong duality holds irrespective of the problem data.

**Theorem 3.5** (Strong duality of (9) and (11)). *We have* $\overline{p}^\star = \underline{d}^\star$.

Theorem 3.5 follows immediately from Sion's classic minimax theorem [45], which applies because $\mathcal{G}_W$ and $\mathcal{G}_V$ are convex as well as compact thanks to [38, Lemma A.6].

By weak duality and the construction of the bounding problems (9) and (11), we trivially have $\underline{d}^\star \leq d^\star \leq p^\star \leq \overline{p}^\star$. Theorem 3.5 reveals that all of these inequalities are in fact equalities, each of which gives rise to a non-trivial insight. The first key insight is that (5) and (6) are strong duals.

**Corollary 2** (Strong duality of (5) and (6)). *We have* $p^\star = d^\star$.

We stress that, unlike Theorem 3.5, Corollary 2 establishes strong duality between two *infinite-dimensional* zero-sum games. The second key implication of Theorem 3.5 is that the distributionally robust LQG problem (5) is solved by a linear output-feedback controller.

**Corollary 3** (The controller's Nash strategy is linear in the observations). *There exist* $U^\star \in \mathcal{U}$ *and* $q^\star \in \mathbb{R}^m$ *such that the distributionally robust LQG problem* (5) *is solved by* $u^\star = q^\star + U^\star y$.

The identity $p^\star = \overline{p}^\star$ readily implies that (5) is solved by a causal controller that is linear in the *purified* observations. However, any causal controller that is linear in the purified observations $\eta$ can be reformulated *exactly* as a causal controller that is linear in the original observations $y$ and vice-versa [6, Proposition 3]. Thus, Corollary 3 follows. The third key implication of Theorem 3.5 is that the *dual* distributionally robust LQG problem is solved by a normal distribution.

**Corollary 4** (Nature's Nash strategy is a normal distribution). *The dual distributionally robust LQG problem* (6) *is solved by a distribution* $\mathbb{P}^\star \in \mathcal{W}_\mathcal{N}$.

Corollary 4 is a direct consequence of the identity $\underline{d}^\star = d^\star$. Note that the optimal normal distribution $\mathbb{P}^\star$ is uniquely determined by the covariance matrices $W^\star$ and $V^\star$ of the exogenous uncertain parameters, which can be computed by solving problem (11). That the worst-case distribution is actually Gaussian is not a-priori expected and is surprising given that the Wasserstein ball contains many non-Gaussian distributions.

## 4. Efficient Numerical Solution of Distributionally Robust LQG Problems

Having proven these structural results, we next turn attention to the problem of finding the optimal strategies. Our next result shows that, under a mild regularity condition, the optimal controller $u^\star$ of the distributionally robust LQG problem (5) can be computed efficiently from $\mathbb{P}^\star$.

**Proposition 4.1** (Optimality of Kalman filter-based feedback controllers). *If* $\hat{V}_t \succ 0$ *for all* $t \in [T-1]$, *then problem* (6) *is solved by a Gaussian distribution* $\mathbb{P}^\star$ *under which* $v_t$ *has a covariance matrix* $V_t^\star \succ 0$ *for every* $t \in [T-1]$, *and* (5) *is solved by the optimal LQG controller corresponding to* $\mathbb{P}^\star$. *Additionally, the optimal value of problem* (9) *and its strong dual* (11) *does not change if we restrict* $\mathcal{G}_W$ *and* $\mathcal{G}_V$ *to* $\mathcal{G}_W^+$ *and* $\mathcal{G}_V^+$, *respectively, where*

$$
\mathcal{G}_W^+ = \left\{ W \in \mathcal{G}_W : X_0 \succeq \lambda_{\min}(\hat{X}_0)I, \ W_t \succeq \lambda_{\min}(\hat{W}_t)I \ \forall t \in [T-1] \right\},
$$

$$
\mathcal{G}_V^+ = \left\{ V \in \mathcal{G}_V : V_t \succeq \lambda_{\min}(\hat{V}_t)I \ \forall t \in [T-1] \right\}.
$$

This implies that the optimal controller can be computed by solving a classic LQG problem corresponding to nature's optimal strategy $\mathbb{P}^\star$, which can be done very efficiently through Kalman filtering and dynamic programming (see Appendix §A for details). It thus suffices to design an efficient algorithm for computing $\mathbb{P}^\star$, which is uniquely determined by the covariance matrices $(W^\star, V^\star)$ that solve problem (11). To this end, we first reformulate (11) as

$$\max_{W \in \mathcal{G}_W^+, V \in \mathcal{G}_V^+} f(W, V), \tag{12}$$

where we restrict $\mathcal{G}_W$ and $\mathcal{G}_V$ to $\mathcal{G}_W^+$ and $\mathcal{G}_V^+$, respectively, due to Proposition 4.1, and where $f(W, V)$ denotes the optimal value function of the inner minimization problem in (11). As (11) is a reformulation of (10) and as the family of all causal purified output-feedback controllers matches the family of causal output-feedback controllers, $f(W, V)$ can also be viewed as the optimal value of the classic LQG problem corresponding to the normal distribution $\mathbb{P}$ determined by the covariance matrices $W$ and $V$. These insights lead to the following structural result.

**Proposition 4.2.** $f(W, V)$ *is concave and $\beta$-smooth in $(W, V) \in \mathcal{G}_W^+ \times \mathcal{G}_V^+$ for some $\beta > 0$.*

By Proposition 4.2, it is possible to address problem (12) with a Frank-Wolfe algorithm [13, 18, 19, 20, 23, 35]. Each iteration of this algorithm solves a direction-finding subproblem, that is, a variant of problem (12) that maximizes the first-order Taylor expansion of $f(W, V)$ around the current iterates.

$$\max_{L_W \in \mathcal{G}_W^+, L_V \in \mathcal{G}_V^+} \langle \nabla_W f(W, V), L_W - W \rangle + \langle \nabla_V f(W, V), L_V - V \rangle \tag{13}$$

The next iterates are then obtained by moving towards a maximizer $(L_W^\star, L_V^\star)$ of (13), i.e., we update

$$(W, V) \leftarrow (W, V) + \alpha \cdot (L_W^\star - W, L_v^\star - V),$$

where $\alpha$ is an appropriate step size. The proposed Frank-Wolfe algorithm enjoys a very low per-iteration complexity because problem (13) is separable. To see this, we reformulate (13) as

$$\max_{L_W, L_V} \ \langle \nabla_{X_0} f(W, V), L_{X_0} - X_0 \rangle + \sum_{t=0}^{T-1} \langle \nabla_{W_t} f(W, V), L_{W_t} - W_t \rangle + \langle \nabla_{V_t} f(W, V), L_{V_t} - V_t \rangle$$
$$\text{s.t.} \quad \mathbb{G}(L_{X_0}, \hat{X}_0)^2 \leq \rho_{x_0}^2, \quad \mathbb{G}(L_{W_t}, \hat{W}_t)^2 \leq \rho_{w_t}^2, \quad \mathbb{G}(L_{V_t}, \hat{V}_t)^2 \leq \rho_{v_t}^2 \quad \forall t \in [T-1]$$
$$L_{X_0} \succeq \lambda_{\min}(\hat{X}_0) I, \qquad L_{W_t} \succeq \lambda_{\min}(\hat{W}_t) I, \qquad L_{V_t} \succeq \lambda_{\min}(\hat{V}_t) I \quad \forall t \in [T-1].$$

Hence, (13) decomposes into $2T + 1$ separate subproblems that can be solved in parallel. That is, for any matrix $Z \in \{X_0, W_0, \ldots, W_{T-1}, V_0, \ldots, V_{T-1}\}$ we solve a separate subproblem of the form

$$\max_{L_Z \succeq \lambda_{\min}(\hat{Z})} \left\{ \langle \nabla_Z f(W, V), L_Z - Z \rangle : \mathbb{G}(L_Z, \hat{Z})^2 \leq \rho_z^2 \right\}. \tag{14}$$

These subproblems can be reformulated as tractable SDPs and are thus amenable to efficient off-the-shelf solvers. By [38, Theorem 6.2], however, one can exploit the structure of the Gelbrich distance in order to reduce (14) to a univariate algebraic equation that can be solved to any desired accuracy $\delta > 0$ by a highly efficient bisection algorithm. We say that $L_Z^\delta$ is a $\delta$-approximate solution of problem (14) for some $\delta \in (0, 1)$ if $L_Z^\delta$ is feasible in (14) and if

$$\langle \nabla_Z f(W, V), L_Z^\delta - Z \rangle \geq \delta \langle \nabla_Z f(W, V), L_Z^\star - Z \rangle,$$

where $L_Z^\star$ is an exact maximizer of (14). Note that, by the concavity of $f(W, V)$, the inner product on the right-hand side is nonnegative and vanishes if and only if $Z$ maximizes $f(W, V)$ over the feasible set of (14). For further details we refer to Appendix §E in the supplementary material.

**Remark 1** (Automatic differentiation). *Recall that $f(W, V)$ coincides with the optimal value of the LQG problem corresponding to the normal distribution $\mathbb{P}$ determined by the covariance matrices $W$ and $V$. By using the underlying dynamic programming equations, $f(W, V)$ can thus be expressed in closed form as a serial composition of $\mathcal{O}(T)$ rational functions (see Appendix §A for details). Hence, $\nabla_Z f(W, V)$ can be calculated symbolically for any $Z \in \{X_0, W_0, \ldots, W_{T-1}, V_0, \ldots, V_{T-1}\}$ by repeatedly applying the chain and product rules. However, the resulting formulas are lengthy and cumbersome. We thus compute the gradients numerically using backpropagation. The cost of evaluating $\nabla_Z f(W, V)$ is then of the same order of magnitude as the cost of evaluating $f(W, V)$.*

A detailed description of the proposed Frank-Wolfe method is given in Algorithm 1 below.

By [31, Theorem 1 and Lemma 7], which applies thanks to the structural properties of $f(W, V)$ established in Proposition 4.2, Algorithm 1 attains a suboptimality gap of $\epsilon$ within $\mathcal{O}(1/\epsilon)$ iterations.

---
**Algorithm 1** Frank-Wolfe algorithm for solving (12)
---
    **Input:** initial iterates $W, V$, nominal covariance matrices $\hat{W}, \hat{V}$, oracle precision $\delta \in (0, 1)$

1: set initial iteration counter $k = 0$
2: **while** stopping criterion is not met **do**
3:     **for** $Z \in \{X_0, W_0, \dots, W_{T-1}, V_0, \dots, V_{T-1}\}$ **do in parallel**
4:         compute $\nabla_Z f(W, V)$
5:         find a $\delta$-approximate solution $L_Z^\delta$ of (14)
6:     **end**
7:     $g \leftarrow \langle \nabla_W f(W, V), L_W^\delta - W \rangle + \langle \nabla_V f(W, V), L_V^\delta - V \rangle$
8:     $(W, V) \leftarrow (W, V) + 2/(2+k) \cdot (L_W^\delta - W, L_V^\delta - V)$
9: **end while**
10: **Output:** $W$ and $V$
---

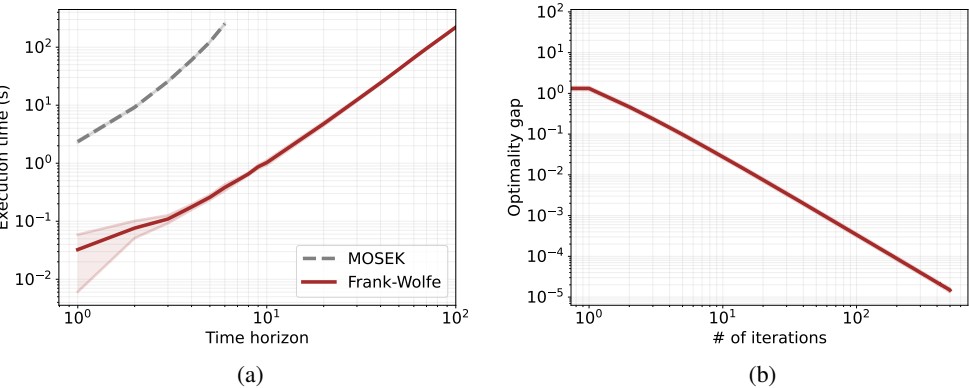

Figure 1: (a) Execution time for MOSEK and Frank-Wolfe algorithm over 10 simulation runs as a function of the horizon $T$ (solid lines show the mean and the shaded areas correspond to 1 standard deviation). (b) Convergence of optimality gap for Frank-Wolfe algorithm with horizon $T = 10$.

## 5. Numerical Experiments

All experiments are run on an Intel i7-8700 CPU (3.2 GHz) machine with 16GB RAM. All linear SDP problems are modeled in Python 3.8.6 using CVXPY [1, 14] and solved with MOSEK [37]. The gradients of $f(W, V)$ are computed via Pymanopt [48] with PyTorch's automated differentiation module [39, 40].

Consider a class of distributionally robust LQG problems with $n = m = p = 10$. We set $A_t = 0.1 \times A$ to have ones on the main diagonal and the superdiagonal and zeroes everywhere else ($A_{i,j} = 1$ if $i = j$ or $i = j - 1$ and $A_{i,j} = 0$ otherwise), and the other matrices to $B_t = C_t = Q_t = R_t = I_d$. The Wasserstein radii are set to $\rho_{x_0} = \rho_{w_t} = \rho_{v_t} = 10^{-1}$. The nominal covariance matrices of the exogenous uncertainties are constructed randomly and with eigenvalues in the interval $[1, 2]$ (so as to ensure they are positive definite). The code is publicly available in the Github repository `https://github.com/RAO-EPFL/DR-Control`.

The optimal value of the distributionally robust LQG problem (5) can be computed by directly solving the SDP reformulation of (11) with MOSEK or by solving the nonlinear SDP (12) with our Frank-Wolfe method detailed in Algorithm 1. We next compare these two approaches in 10 independent simulation runs, where we set a stopping criterion corresponding to an optimality gap below $10^{-3}$ and we run the Frank-Wolfe method with $\delta = 0.95$. Figure 1a illustrates the execution time for both approaches as a function of the planning horizon $T$; runs where MOSEK exceeds 100s are not reported. Figure 1b visualizes the empirical convergence behavior of the Frank-Wolfe algorithm. The results highlight that the Frank-Wolfe algorithm achieves running times that are uniformly lower than MOSEK across all problem horizons and is able to find highly accurate solutions already after a small number of iterations (50 iterations for problem instances of time horizon $T = 10$).

# 6. Concluding Remarks and Limitations

In view of the popularity of LQG models, the results in this work carry important theoretical and practical implications. Despite considering a generalization of the classic LQG setting where the noise affecting the system dynamics and the observations follows unknown (and potentially non-Gaussian) distributions, our findings suggest that certain classic structural results continue to hold and that highly efficient methods can be adapted to tackle this more realistic (and more challenging) problem. Specifically, that control policies depending linearly on observations continue to be optimal and that the worst-case distribution turns out to be Gaussian is surprising from a theoretical angle and also has direct practical implications, because it allows leveraging the highly efficient Kalman filter in conjunction with dynamic programming and a Frank-Wolfe method to design an efficient computational procedure for solving the problem.

The results also raise several important questions that warrant future exploration. First, it would be highly relevant to consider extensions where the system matrices are also affected by uncertainty, as this captures many applications of practical interest in, e.g., reinforcement learning or revenue management. Second, it would be worth exploring an infinite horizon setting or relaxing the assumption that the nominal distribution is Gaussian, as both assumptions may be limiting the practical appeal of the framework. Third, one could also attempt to prove structural optimality results or design novel algorithms for generating high-quality suboptimal solutions for the more general setting involving constraints on states and/or control inputs. Lastly, one could improve the present algorithmic proposal by exploiting topological properties of the objective so as to guarantee linear convergence rates in the Frank-Wolfe procedure.

**Acknowledgements.** This research was supported by the Swiss National Science Foundation under the NCCR Automation, grant agreement 51NF40_180545. Dan A. Iancu would like to acknowledge INSEAD for financial support during the duration of the project.

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

## Appendix

The supplementary material is structured as follows. Appendix §A presents the well-known solution to the classic LQG problem using dynamic programming and Kalman Filter estimation. Appendix §B provides the definitions of the stacked system matrices utilized in the compact formulation (5) of the distributionally robust LQG problem. Appendix §C contains the proofs of the formal statements in the main text and provides additional technical results. Appendix §D derives the SDP reformulation of the dual problem (11). Appendix §E, finally, elaborates on the bisection algorithm used for solving the linearization oracle of the Frank-Wolfe algorithm.

## A. Solution of the LQG Problem

The classic LQG problem can be solved efficiently via dynamic programming; see, e.g., [8]. That is, the unique optimal control inputs satisfy $u_t^\star = K_t \hat{x}_t$ for every $t \in [T-1]$, where $K_t \in \mathbb{R}^{n \times n}$ is the optimal feedback gain matrix, and $\hat{x}_t = \mathbb{E}_\mathbb{P}[x_t | y_0, \ldots, y_t]$ is the minimum mean-squared-error estimator of $x_t$ given the observation history up to time $t$. Thanks to the celebrated separation principle, $K_t$ can be computed by pretending that the system is deterministic and allows for perfect state observations, and $\hat{x}_t$ can be computed while ignoring the control problem.

To compute $K_t$, one first solves the deterministic LQR problem corresponding to the LQG problem at hand. Its value function $x_t^\top P_t x_t$ at time $t$ is quadratic in $x_t$, and $P_t$ obeys the backward recursion

$$P_t = A_t^\top P_{t+1} A_t + Q_t - A_t^\top P_{t+1} B_t (R_t + B_t^\top P_{t+1} B_t)^{-1} B_t^\top P_{t+1} A_t \quad \forall t \in [T-1] \quad \text{(A.15a)}$$

initialized by $P_T = Q_T$. The optimal feedback gain matrix $K_t$ can then be computed from $P_{t+1}$ as

$$K_t = -(R_t + B_t^\top P_{t+1} B_t)^{-1} B_t^\top P_{t+1} A_t \quad \forall t \in [T-1]. \quad \text{(A.15b)}$$

Since $x_t$ and $(y_0, \ldots, y_t)$ are jointly normally distributed, the minimum mean-squared-error estimator $\hat{x}_t$ can be calculated directly using the formula for the mean of a conditional normal distribution. Alternatively, however, one can use the Kalman filter to compute $\hat{x}_t$ recursively, which is significantly more insightful and efficient. The Kalman filter also recursively computes the covariance matrix $\Sigma_t$ of $x_t$ conditional on $y_0, \ldots, y_t$ and the covariance matrix $\Sigma_{t+1|t}$ of $x_{t+1}$ conditional on $y_0, \ldots, y_t$ evaluated under $\mathbb{P}$. Specifically, these covariance matrices obey the forward recursion

$$\left. \begin{aligned} \Sigma_t &= \Sigma_{t|t-1} - \Sigma_{t|t-1} C_t^\top (C_t \Sigma_{t|t-1} C_t^\top + V_t)^{-1} C_t \Sigma_{t|t-1} \\ \Sigma_{t+1|t} &= A_t \Sigma_t A_t^\top + W_t \end{aligned} \right\} \quad \forall t \in [T-1] \quad \text{(A.16)}$$

initialized by $\Sigma_{0|-1} = X_0$. Using $\Sigma_{t|t-1}$, we then define the Kalman filter gain as

$$L_t = \Sigma_t C_t^\top V_t^{-1} \quad \forall t \in [T-1]$$

which allows us to compute the minimum mean-squared-error estimator via the forward recursion

$$\hat{x}_{t+1} = A_t \hat{x}_t + B_t u_t + L_{t+1} (y_{t+1} - C_{t+1}(A_t \hat{x}_t + B_t u_t)) \quad \forall t \in [T-1]$$

initialized by $\hat{x}_0 = L_0 y_0$. One can also show that the optimal value of the LQG problem amounts to

$$\sum_{t=0}^{T-1} \text{Tr}((Q_t - P_t)\Sigma_t) + \sum_{t=1}^{T} \text{Tr}(P_t(A_{t-1}\Sigma_{t-1}A_{t-1}^\top + W_{t-1})) + \text{Tr}(P_0 X_0). \quad \text{(A.17)}$$

## B. Definitions of Stacked System Matrices

The stacked system matrices appearing in the distributionally robust LQG problem (5) are defined as follows. First, the stacked state and input cost matrices $Q \in \mathbb{S}^{n(T+1)}$ and $R \in \mathbb{S}^{mT}$ are set to

$$Q = \begin{bmatrix} Q_0 & & & \\ & Q_1 & & \\ & & \ddots & \\ & & & Q_T \end{bmatrix} \quad \text{and} \quad R = \begin{bmatrix} R_0 & & & \\ & R_1 & & \\ & & \ddots & \\ & & & R_{T-1} \end{bmatrix},$$

respectively. Similarly, the stacked matrices appearing in the linear dynamics and the measurement equations $C \in \mathbb{R}^{pT \times n(T+1)}$, $G \in \mathbb{R}^{n(T+1) \times n(T+1)}$ and $H \in \mathbb{R}^{n(T+1) \times mT}$ are defined as

$$
C = \begin{bmatrix} C_0 & 0 & & \\ & C_1 & 0 & \\ & & \ddots & \ddots \\ & & & C_{T-1} & 0 \end{bmatrix}, \quad G = \begin{bmatrix} A_0^0 & & & \\ A_0^1 & A_1^1 & & \\ \vdots & & \ddots & \\ A_0^T & A_1^T & \dots & A_T^T \end{bmatrix}
$$

and

$$
H = \begin{bmatrix} 0 & & & & \\ A_1^1 B_0 & 0 & & & \\ A_1^2 B_0 & A_2^2 B_1 & 0 & & \\ \vdots & & & \ddots & \\ \vdots & & & & 0 \\ A_1^T B_0 & A_2^T B_1 & \dots & \dots & A_T^T B_{T-1} \end{bmatrix},
$$

respectively, where $A_s^t = \prod_{k=s}^{t-1} A_k$ for every $s < t$ and $A_s^t = I_n$ for $s = t$.

Using the stacked system matrices, we can now express the purified observation process $\eta$ as a linear function of the exogenous uncertainties $w$ and $v$ that is *not* impacted by $u$; see also [5, 46]

**Lemma B.1.** *We have $\eta = Dw + v$, where $D = CG$.*

*Proof of Lemma B.1.* The purified observation process is defined as $\eta = y - \hat{y}$. Recall now that the observations of the original system satisfy $y = Cx + v$. Similarly, one readily verifies that the observations of the fictitious noise-free system satisfy $\hat{y} = C\hat{x}$. Thus, we have $\eta = C(x - \hat{x}) + v$. Next, recall that the state of the original system satisfies $x = Hu + Gw$, and note that the state of the fictitious noise-free system satisfies $\hat{x} = Hu$. Combining all of these linear equations finally shows that $u$ cancels out and that $\eta = CGw + v = Dw + v$. □

## C. Proofs

### C.1. Additional Technical Results

It is well known that every causal controller that is linear in the original observations $y$ can be reformulated as a causal controller that is linear in the purified observations $\eta$ and vice versa [5, 46]. Perhaps surprisingly, however, the one-to-one transformation between the respective coefficients of $y$ and $\eta$ is *not* linear. To keep this paper self-contained, we review these insights in the next lemma.

**Lemma C.1.** *If $u = U\eta + q$ for some $U \in \mathcal{U}$ and $q \in \mathbb{R}^{pT}$, then $u = U'y + q'$ for $U' = (I + UCH)^{-1}U$ and $q' = (I + UCH)^{-1}q$. Conversely, if $u = U'y + q'$ for some $U' \in \mathcal{U}$ and $q' \in \mathbb{R}^{pT}$, then $u = U\eta + q$ for $U = (I - U'CH)^{-1}U'$ and $q = (I - U'CH)^{-1}q'$.*

*Proof of Lemma C.1.* If $u = U\eta + q$ for some $U \in \mathcal{U}$ and $q \in \mathbb{R}^{pT}$, then we have

$$
u = U\eta + q = U(y - \hat{y}) + q = Uy - UC\hat{x} + q = Uy - UCHu + q,
$$

where the second equality follows from the definition of $\eta$, the third equality holds because $y = Cx + v$, and the last equality exploits our earlier insight that $\hat{y} = C\hat{x}$. The last expression depends only on $y$ and $u$. Solving for $u$ yields $u = U'y + q'$, where $U' = (I + UCH)^{-1}U$ and $q' = (I + UCH)^{-1}q$. Note that $(I + UCH)$ is indeed invertible because $I + UCH$ is a lower triangular matrix with all diagonal entries equal to one, ensuring a determinant of one.

Similarly, if $u = U'y + q'$ for some $U' \in \mathcal{U}$ and $q' \in \mathbb{R}^{pT}$, then we have

$$
u = U'y + q' = U'(\eta + \hat{y}) + q' = U'\eta + U'C\hat{x} + q' = U'\eta + U'CHu + q'.
$$

Solving for $u$ yields $u = U\eta + q$, where $U = (I - U'CH)^{-1}U'$ and $q = (I - U'CH)^{-1}q'$. Note again that $(I - U'CH)$ is indeed invertible because $(I - U'CH)$ is a lower triangular matrix with all diagonal entries equal to one. □

**C.2. Proofs of Section 3**

*Proof of Proposition 3.2.* In problem (8), both $u$ and $x$ are linear in $w$ and $v$, i.e., $u = q + UDw + Uv$ and $x = Hu + Gw = Hq + HUDw + HUv + Gw$. By substituting the linear representations of $u$ and $x$ into the objective function of problem (8), we obtain the following equivalent reformulation.

$$\min_{\substack{q \in \mathbb{R}^{pT} \\ U \in \mathcal{U}}} \max_{\mathbb{P} \in \mathcal{G}} \mathbb{E}_{\mathbb{P}} \left[ w^\top \left( D^\top U^\top (R + H^\top QH)UD + 2D^\top U^\top H^\top QG + G^\top QG \right) w \right]$$
$$+ \mathbb{E}_{\mathbb{P}} \left[ v^\top \left( U^\top (R + H^\top QH)U \right) v \right] + q^\top (R + H^\top QH)q$$

For any fixed $\mathbb{P} \in \mathcal{G}$, we can express the expectation in the objective function of the above problem in terms of the covariance matrices $W = \mathbb{E}_{\mathbb{P}}[ww^\top]$ and $V = \mathbb{E}_{\mathbb{P}}[vv^\top]$. Thus, the problem becomes

$$\min_{\substack{q \in \mathbb{R}^{pT} \\ U \in \mathcal{U}}} \max_{W, V, \mathbb{P}} \quad \mathrm{Tr}\left( (D^\top U^\top (R + H^\top QH)UD + 2G^\top QHUD + G^\top QG)W \right)$$
$$+ \mathrm{Tr}\left( (U^\top (R + H^\top QH)U)V \right) + q^\top (R + H^\top QH)q \qquad \text{(A.18)}$$
$$\text{s.t.} \quad \mathbb{P} \in \mathcal{G}, \quad W = \mathbb{E}_{\mathbb{P}}[ww^\top], \quad V = \mathbb{E}_{\mathbb{P}}[vv^\top].$$

Recall now the definition of $\mathcal{G}$, and note that the requirements $\mathbb{G}(X_0, \hat{X}_0) \le \rho_{x_0}$, $\mathbb{G}(W_t, \hat{W}_t) \le \rho_{w_t}$ and $\mathbb{G}(V_t, \hat{V}_t) \le \rho_{v_t}$ are equivalent to the convex constraints $\mathbb{G}(X_0, \hat{X}_0)^2 \le \rho_{x_0}^2$, $\mathbb{G}(W_t, \hat{W}_t)^2 \le \rho_{w_t}^2$ and $\mathbb{G}(V_t, \hat{V}_t)^2 \le \rho_{v_t}^2$, respectively, for all $t \in [T-1]$. The definition of $\mathcal{G}$ also implies that

$$W = \mathbb{E}_{\mathbb{P}}[ww^\top] = \mathrm{diag}(X_0, W_0, \ldots, W_{T-1}) \quad \text{and} \quad V = \mathbb{E}_{\mathbb{P}}[vv^\top] = \mathrm{diag}(V_0, \ldots, V_{T-1}).$$

Problem (A.18) thus constitutes a relaxation of problem (9). Indeed, the feasible set of the inner maximization problem in (A.18) is a subset of the feasible set of the inner maximization problem in (9). Moreover, for any $W$ and $V$ feasible in the inner maximization problem in (9), the distribution $\mathbb{P} = \mathbb{P}_{x_0} \otimes (\otimes_{t=0}^{T-1} \mathbb{P}_{w_t}) \otimes (\otimes_{t=0}^{T} \mathbb{P}_{v_t})$ defined through $\mathbb{P}_{x_0} = \mathcal{N}(0, X_0)$, $\mathbb{P}_{w_t} = \mathcal{N}(0, W_t)$ and $\mathbb{P}_{v_t} = \mathcal{N}(0, V_t)$, $t \in [T-1]$, is feasible in the inner maximization problem in (A.18) with the same objective value. The relaxation is thus exact, and the optimal values of (8), (9) and (A.18) coincide. $\qquad \square$

*Proof of Proposition 3.4.* Recall that the space $\mathcal{U}_y$ of all causal output-feedback controllers coincides with the space $\mathcal{U}_\eta$ of all causal *purified* output-feedback controllers. We can thus replace the feasible set $\mathcal{U}_\eta$ of the inner minimization problem in (10) with $\mathcal{U}_y$. Hence, for any fixed $\mathbb{P} \in \mathcal{W}_\mathcal{N}$, the inner minimization problem in (10) constitutes a classic LQG problem. By standard LQG theory [8], it is solved by a *linear* output-feedback controller of the form $u = U'y + q'$ for some $U' \in \mathcal{U}$ and $q' \in \mathbb{R}^{pT}$; see also Appendix §A. Lemma C.1 shows, however, that any linear output-feedback controller can be equivalently expressed as a linear *purified*-output feedback controller of the form $u = U\eta + q$ for some $U \in \mathcal{U}$ and $q \in \mathbb{R}^{pT}$. In summary, the above reasoning shows that the feasible set of the inner minimization problem in (10) can be reduced to the family of all linear purified-output feedback controllers without sacrificing optimality. Thus, problem (10) is equivalent to

$$\max_{\mathbb{P} \in \mathcal{W}_\mathcal{N}} \min_{q, U, x, u} \quad \mathbb{E}_{\mathbb{P}} \left[ u^\top Ru + x^\top Qx \right]$$
$$\text{s.t.} \quad U \in \mathcal{U}, \quad u = q + U\eta, \quad x = Hu + Gw.$$

Using a similar reasoning as in the proof of Proposition 3.2, we can now substitute the linear representations of $u$ and $x$ into the objective function and reformulate the above problem as

$$\max_{W, V, \mathbb{P}} \min_{\substack{q \in \mathbb{R}^{pT} \\ U \in \mathcal{U}}} \quad \mathrm{Tr}\left( (D^\top U^\top (R + H^\top QH)UD + 2G^\top QHUD + G^\top QG)W \right)$$
$$+ \mathrm{Tr}\left( (U^\top (R + H^\top QH)U)V \right) + q^\top (R + H^\top QH)q$$
$$\text{s.t.} \quad \mathbb{P} \in \mathcal{W}_\mathcal{N}, \quad W = \mathbb{E}_{\mathbb{P}}[ww^\top], \quad V = \mathbb{E}_{\mathbb{P}}[vv^\top].$$

As $\mathcal{W}_\mathcal{N}$ contains only *normal* distributions, Proposition 3.3 implies that $\mathbb{W}(\mathbb{P}_{x_0}, \hat{\mathbb{P}}_{x_0}) = \mathbb{G}(X_0, \hat{X}_0)$, $\mathbb{W}(\mathbb{P}_{w_t}, \hat{\mathbb{P}}_{w_t}) = \mathbb{G}(W_t, \hat{W}_t)$ and $\mathbb{W}(\mathbb{P}_{v_t}, \hat{\mathbb{P}}_{v_t}) = \mathbb{G}(V_t, \hat{V}_t)$ for all $t \in [T-1]$. We may thus replace the requirement $\mathbb{W}(\mathbb{P}_{x_0}, \hat{\mathbb{P}}_{x_0}) \le \rho_{x_0}$ in the definition of $\mathcal{W}_\mathcal{N}$ by $\mathbb{G}(X_0, \hat{X}_0) \le \rho_{x_0}$, which is equivalent to the convex constraint $\mathbb{G}(X_0, \hat{X}_0)^2 \le \rho_{x_0}^2$. The conditions on the marginal distributions of $w_t$ and $v_t$, $t \in [T-1]$, admit similar reformulations. The definition of $\mathcal{W}_\mathcal{N}$ also implies that

$$W = \mathbb{E}_{\mathbb{P}}[ww^\top] = \mathrm{diag}(X_0, W_0, \ldots, W_{T-1}) \quad \text{and} \quad V = \mathbb{E}_{\mathbb{P}}[vv^\top] = \mathrm{diag}(V_0, \ldots, V_{T-1}).$$

Thus, the feasible set of the outer maximization problem in (11) constitutes a relaxation of that in (10). One readily verifies that the relaxation is exact by using similar arguments as in the proof of Proposition 3.2. Thus, the claim follows. □

*Proof of Theorem 3.5.* By Proposition 3.2, $\bar{p}^\star$ coincides with the minimum of (9). Similarly, by Proposition 3.4 $\underline{d}^\star$ coincides with the maximum of (11). Note that problems (9) and (11) only differ by the order of minimization and maximization. Note also that $\mathcal{U}$ is convex and closed, $\mathcal{G}_W$ and $\mathcal{G}_V$ are convex and compact by virtue of [38, Lemma A.6], and the (identical) trace terms in (9) and (11) are bilinear in $(W, V)$ and $(U, q)$. The claim thus follows from Sion's minimax theorem [45]. □

## C.3. Proofs of Section 4

Note that Proposition 4.1 is consistent with Corollary 3 because the optimal LQG controller corresponding to $\mathbb{P}^\star$ is linear in the past observations.

*Proof of Proposition 4.1.* By [38, Lemma A.3], the inner problem in (9) admits a maximizer $(W^\star, V^\star)$ with $X_0^\star \succeq \lambda_{\min}(\hat{X}_0)$ as well as $W_t^\star \succeq \lambda_{\min}(\hat{W}_t)$ and $V_t^\star \succeq \lambda_{\min}(\hat{V}_t)$ for all $t \in [T-1]$. Thus, the optimal value of problem (9) and its strong dual (11) does not change if we restrict $\mathcal{G}_W$ and $\mathcal{G}_V$ to $\mathcal{G}_W^+$ and $\mathcal{G}_V^+$, respectively. We may thus conclude that problem (11) has a maximizer $(W^\star, V^\star)$ with $V_t^\star \succeq \lambda_{\min}(\hat{V}_t) \succ 0$ for all $t \in [T-1]$. This in turn implies that problem (6) is solved by a normal distribution $\mathbb{P}^\star$ under which the covariance matrix of the observation noise $v_t$ satisfies $V_t^\star \succ 0$ for every $t \in [T-1]$. As (5) and (6) are strong duals, the optimal solution $u^\star$ of problem (5) forms a Nash equilibrium with $\mathbb{P}^\star$, i.e., $u^\star$ is a best response to $\mathbb{P}^\star$ and thus solves the *classic* LQG problem corresponding to $\mathbb{P}^\star$. As $R_t \succ 0$ for every $t \in [T-1]$, this best response $u^\star$ is unique, and as $V_T^\star \succ 0$ for every $t \in [T-1]$, $u^\star$ is in fact the Kalman filter-based optimal output-feedback strategy corresponding to $\mathbb{P}^\star$ (which can be obtained using the techniques highlighted in Appendix §A). □

Before proving Proposition 4.2, recall that $f(W, V)$ is called $\beta$-smooth for some $\beta > 0$ if

$$|\nabla f(W, V) - \nabla f(W', V')| \leq \beta \left( \|W - W'\|_F^2 + \|V - V'\|_F^2 \right)^{\frac{1}{2}} \quad \forall W, W' \in \mathcal{G}_W^+, \ V, V' \in \mathcal{G}_V^+,$$

where $\|\cdot\|_F$ denotes the Frobenius norm.

*Proof of Proposition 4.2.* The function $f(W, V)$ is concave because the objective function of the inner minimization problem in (11) is linear (and hence concave) in $W$ and $V$ and because concavity is preserved under minimization. To prove that $f(W, V)$ is $\beta$-smooth, we first recall from Proposition 3.3 that it coincides with the optimal value of the inner minimization problem in (10). As $\mathcal{U}_\eta = \mathcal{U}_y$, $f(W, V)$ can thus be viewed as the optimal value of the classic LQG problem corresponding to the normal distribution $\mathbb{P}$ determined by the covariance matrices $W$ and $V$. Hence, $f(W, V)$ coincides with (A.17), where $\Sigma_t$, for $t \in [T-1]$, is a function of $(W, V)$ defined recursively through the Kalman filter equations (A.16). Note that all inverse matrices in (A.16) are well-defined because any $V \in \mathcal{G}_V^+$ is strictly positive definite. Therefore, $\Sigma_t$ constitutes a proper rational function (that is, a ratio of two polynomials with the polynomial in the denominator being strictly positive) for every $t \in [T-1]$. Thus, $f(W, V)$ is infinitely often continuously differentiable on a neighborhood of $\mathcal{G}_W^+ \times \mathcal{G}_V^+$.

As $f(W, V)$ is concave and (at least) twice continuously differentiable, it is $\beta$-smooth on $\mathcal{G}_W^+ \times \mathcal{G}_V^+$ if and only if the largest eigenvalue of the Hessian matrix of $-f(W, V)$ is bounded above by $\beta$ throughout $\mathcal{G}_W^+ \times \mathcal{G}_V^+$. Also, the largest eigenvalue of the positive semidefinite Hessian matrix $\nabla^2(-f(W, V))$ coincides with the spectral norm of $\nabla^2 f(W, V)$. We may thus set

$$\beta = \sup_{W \in \mathcal{G}_W^+, V \in \mathcal{G}_V^+} \|\nabla^2 f(W, V)\|_2, \tag{A.19}$$

where $\|\cdot\|_2$ denotes the spectral norm. The supremum in the above maximization problem is finite and attained thanks to Weierstrass' theorem, which applies because $f(W, V)$ is twice continuously differentiable and the spectral norm is continuous, while the sets $\mathcal{G}_W^+$ and $\mathcal{G}_V^+$ are compact by virtue of [38, Lemma A.6]. This observation completes the proof. □

## D. SDP Reformulation of the Lower Problem (11)

Instead of solving the dual problem (11) with the customized Frank-Wolfe algorithm of Section 4, it can be reformulated as an SDP amenable to off-the-shelf solvers. This reformulation is obtained by dualizing the inner minimization problem and by exploiting the following preliminary lemma.

**Lemma D.1.** *For any $\hat{Z} \in \mathbb{S}_+^d$ and $\rho_z \geq 0$, the set $\mathcal{G}_Z = \{Z \in \mathbb{S}_+^d : \mathbb{G}(Z, \hat{Z}) \leq \rho_z\}$ coincides with*

$$\left\{ Z \in \mathbb{S}_+^d : \exists E_z \in \mathbb{S}_+^d \text{ with } \mathrm{Tr}(Z + \hat{Z} - 2E_z) \leq \rho_z^2, \begin{bmatrix} \hat{Z}^{\frac{1}{2}} Z \hat{Z}^{\frac{1}{2}} & E_z \\ E_z & I \end{bmatrix} \succeq 0 \right\}.$$

*Proof of Lemma D.1.* By Definition 2, we have

$$\mathcal{G}_Z = \{Z \in \mathbb{S}_+^d : \mathrm{Tr}(Z + \hat{Z} - 2(\hat{Z}^{\frac{1}{2}} Z \hat{Z}^{\frac{1}{2}})^{\frac{1}{2}}) \leq \rho_z^2\}.$$

Next, introduce an auxiliary variable $E_z \in \mathbb{S}_+^d$ subject to the matrix inequality $E_z^2 \preceq (\hat{Z}^{\frac{1}{2}} Z \hat{Z}^{\frac{1}{2}})$. By [4, Theorem 1], this inequality can be recast as $E_z \preceq (\hat{Z}^{\frac{1}{2}} Z \hat{Z}^{\frac{1}{2}})^{\frac{1}{2}}$. Hence, we can reformulate the nonlinear matrix inequality in the above representation of $\mathcal{G}_Z$ as $\mathrm{Tr}(Z + \hat{Z} - 2E_z) \leq \rho_z^2$. A standard Schur complement argument reveals that the inequality $E_z^2 \preceq (\hat{Z}^{\frac{1}{2}} Z \hat{Z}^{\frac{1}{2}})$ is also equivalent to

$$\begin{bmatrix} \hat{Z}^{\frac{1}{2}} Z \hat{Z}^{\frac{1}{2}} & E_z \\ E_z & I \end{bmatrix} \succeq 0.$$

The claim then follows by combining all of these insights. $\qquad\square$

We are now ready to derive the desired SDP reformulation of problem (11).

**Proposition D.2.** *If $\hat{V} \succ 0$, then problem (11) is equivalent to the SDP*

$$
\begin{aligned}
\max \quad & \mathrm{Tr}(G^\top Q G W) - \mathrm{Tr}(F(R + H^\top Q H)^{-1}) \\
\text{s.t.} \quad & W \in \mathbb{S}_+^{n(T+1)}, \ V \in \mathbb{S}_+^{pT}, \ M \in \mathcal{M}, \ F \in \mathbb{S}_+^{Tm} \\
& E_{x_0} \in \mathbb{S}_+^n, \ E_{w_t} \in \mathbb{S}_+^n, \ E_{v_t} \in \mathbb{S}_+^p \quad \forall t \in [T-1] \\
& \mathrm{Tr}(W_0 + \hat{X}_0 - 2E_{x_0}) \leq \rho_{x_0}^2, \\
& \mathrm{Tr}(W_{t+1} + \hat{W}_t - 2E_{w_t}) \leq \rho_{w_t}^2, \ \mathrm{Tr}(V_t + \hat{V}_t - 2E_{v_t}) \leq \rho_{v_t}^2 \quad \forall t \in [T-1] \\
& \begin{bmatrix} \hat{X}_0^{\frac{1}{2}} X_0 \hat{X}_0^{\frac{1}{2}} & E_{x_0} \\ E_{x_0} & I_n \end{bmatrix} \succeq 0, \\
& \begin{bmatrix} \hat{W}_t^{\frac{1}{2}} W_{t+1} \hat{W}_t^{\frac{1}{2}} & E_{w_t} \\ E_{w_t} & I_n \end{bmatrix} \succeq 0, \ \begin{bmatrix} \hat{V}_t^{\frac{1}{2}} V_t \hat{V}_t^{\frac{1}{2}} & E_{v_t} \\ E_{v_t} & I_p \end{bmatrix} \succeq 0 \quad \forall t \in [T-1] \\
& \begin{bmatrix} F & H^\top Q G W D^\top + M/2 \\ (H^\top Q G W D^\top + M/2)^\top & D W D^\top + V \end{bmatrix} \succeq 0 \\
& W_0 \succeq \lambda_{\min}(\hat{X}_0)I, \quad W_{t+1} \succeq \lambda_{\min}(\hat{W}_t)I, \quad V_t \succeq \lambda_{\min}(\hat{V}_t)I \quad \forall t \in [T-1].
\end{aligned}
\tag{A.20}
$$

*Here, $\mathcal{M}$ denotes the set of all strictly upper block triangular matrices of the form*

$$\begin{bmatrix} 0 & M_{1,2} & M_{1,3} & \cdots & M_{1,T} \\ & 0 & M_{2,3} & & M_{2,T} \\ & & \ddots & & \vdots \\ & & & 0 & M_{T-1,T} \\ & & & & 0 \end{bmatrix} \in \mathbb{R}^{Tm \times Tp},$$

*where $M_{t,s} \in \mathbb{R}^{m \times p}$ for every $t, s \in \mathbb{Z}$ with $1 \leq t < s \leq T$.*

*Proof of Proposition D.2.* The proof relies on dualizing the inner minimization problem in (11). Note that strong duality holds because the primal problem is trivially feasible and involves only equality constraints, which implies that any feasible point is in fact a Slater point. In the following we use $M \in \mathcal{M}$ to denote the Lagrange multiplier of the constraint $U \in \mathcal{U}$, which requires all blocks of

the matrix $U$ above the main diagonal to vanish. The Lagrangian function of the inner minimization problem in (11) can therefore be represented as

$$\mathcal{L}(q, U, M) = \mathrm{Tr}\left( \left( D^\top U^\top (R + H^\top QH) UD + G^\top QG \right) W \right) + 2\,\mathrm{Tr}(G^\top QHUDW)$$
$$+ \mathrm{Tr}\left( \left( U^\top (R + H^\top QH) U \right) V \right) + q^\top (R + H^\top QH) q + \mathrm{Tr}(UM^\top).$$

Recall now that $R \succ 0$ and $Q \succeq 0$, and thus $R + H^\top QH \succ 0$. Consequently, $\mathcal{L}$ is minimized by $q^\star = 0$ for any fixed $U$ and $M$. In addition, the partial gradient of $\mathcal{L}$ with respect $U$ is given by

$$\frac{\partial \mathcal{L}}{\partial U} = 2(R + H^\top QH) UDWD^\top + 2(R + H^\top QH)UV + 2H^\top QGWD^\top + M.$$

Recall also that $V \in \mathcal{G}_V^+$ is strictly positive, which implies that $DWD^\top + V \succ 0$ is invertible. As we already know that $R + H^\top QH \succ 0$ is invertible, as well, $\mathcal{L}$ is minimized by

$$U^\star = -(R + H^\top QH)^{-1} \left( H^\top QGWD^\top + M/2 \right) (DWD^\top + V)^{-1}$$

for any fixed $M$. Substituting both $q^\star$ and $U^\star$ into $\mathcal{L}$ yields the dual objective function

$$g(M) = \mathcal{L}(q^\star, U^\star, M) = \mathrm{Tr}(G^\top QGW)$$
$$- \mathrm{Tr}\left( (R + H^\top QH)^{-1} (H^\top QGWD^\top + M/2)(DWD^\top + V)^{-1}(H^\top QGWD^\top + M/2)^\top \right).$$

The dual of the inner minimization problem in (11) is thus given by $\max_{M \in \mathcal{M}} g(M)$. To linearize the dual objective function, we next introduce an auxiliary variable $F \in \mathbb{S}_+^{mT}$ subject to the matrix inequality $F \succeq (H^\top QGWD^\top + M/2)(DWD^\top + V)^{-1}(H^\top QGWD^\top + M/2)^\top$. By using a standard Schur complement reformulation, we can then rewrite the dual problem as

$$\begin{aligned}
\max \quad & \mathrm{Tr}(G^\top QGW) - \mathrm{Tr}((R + H^\top QH)^{-1} F) \\
\mathrm{s.t.} \quad & M \in \mathcal{M},\ F \in \mathbb{S}_+^{mT} \\
& \begin{bmatrix} F & H^\top QGWD^\top + M/2 \\ (H^\top QGWD^\top + M/2)^\top & DWD^\top + V \end{bmatrix} \succeq 0.
\end{aligned} \tag{A.21}$$

Next, by replacing the inner problem in (11) with its strong dual (A.21), we can reformulate (11) as

$$\begin{aligned}
\max \quad & \mathrm{Tr}(G^\top QGW) - \mathrm{Tr}((R + H^\top QH)^{-1} F) \\
\mathrm{s.t.} \quad & M \in \mathcal{M},\ F \in \mathbb{S}_+^{mT},\ W \in \mathbb{S}_+^{n(T+1)},\ V \in \mathbb{S}_+^{pT} \\
& \begin{bmatrix} F & H^\top QGWD^\top + M/2 \\ (H^\top QGWD^\top + M/2)^\top & DWD^\top + V \end{bmatrix} \succeq 0 \\
& \mathbb{G}(X_0, \hat{X}_0)^2 \leq \rho_{x_0}^2,\ \mathbb{G}(W_t, \hat{W}_t) \leq \rho_{w_t}^2,\ \mathbb{G}(V_t, \hat{V}_t) \leq \rho_{v_t}^2 \quad \forall t \in [T-1].
\end{aligned} \tag{A.22}$$

By Proposition 4.1, the inclusion of the constraints $X_0 \succeq \lambda_{\min}(\hat{X}_0)I$, $W_t \succeq \lambda_{\min}(\hat{W}_t)I$ and $V_t \succeq \lambda_{\min}(\hat{V}_t)I$ for all $t \in [T-1]$ has no effect on the solution to problem (A.22). In addition, by Lemma D.1, each (non-linear) Gelbrich constraint in (A.22) can be reformulated as an equivalent (linear) SDP constraint. Thus, problem (A.22) reduces to (A.20), and the claim follows. $\qquad\square$

## E. Bisection Algorithm for the Linearization Oracle

We now show that the direction-finding subproblem (14) can be solved efficiently via bisection. To this end, we first establish that (14) can be reduced to the solution of a univariate algebraic equation.

**Proposition E.1** ([38, Proposition A.4 (iii)]). *If $\hat{Z} \in \mathbb{S}_{++}^d$, $\Gamma_Z \in \mathbb{S}_+^d$, $\Gamma_Z \neq 0$ and $\rho_z \in \mathbb{R}_{++}$, then*

$$\begin{aligned}
\max \quad & \langle \Gamma_Z, L - Z \rangle \\
\mathrm{s.t.} \quad & \mathbb{G}(L, \hat{Z}) \leq \rho_z \\
& L \succeq \lambda_{\min}(\hat{Z})I
\end{aligned} \tag{A.23}$$

*is uniquely solved by $L^\star = (\gamma^\star)^2 (\gamma^\star I - \Gamma_Z)^{-1} \hat{Z} (\gamma^\star I - \Gamma_Z)^{-1}$, where $\gamma^\star$ is the unique solution of*

$$\rho_z^2 - \langle \hat{Z}, (I - \gamma^\star(\gamma^\star I - \Gamma_Z)^{-1})^2 \rangle = 0 \tag{A.24}$$

*in the interval $(\lambda_{\max}(\Gamma_Z), \infty)$.*

In practice, we need to solve the algebraic equation (A.24) numerically. The numerical error in approximating $\gamma^\star$ should be contained to ensure that $L^\star$ approximates the exact maximizer of problem (A.23). The next proposition shows that, for any tolerance $\delta \in (0,1)$, a $\delta$-approximate solution of (A.23) can be computed with an efficient bisection algorithm.

**Proposition E.2** ([38, Theorem 6.4]). *For any fixed $\rho_z \in \mathbb{R}_{++}$, $\hat{Z} \in \mathbb{S}_{++}^d$ and $\Gamma_Z \in \mathbb{S}_+^d, \Gamma_Z \neq 0$, define $\mathcal{G}_Z^+ = \{Z \in \mathbb{S}_+^d : \mathbb{G}(Z, \hat{Z}) \leq \rho_z, Z \succeq \lambda_{\min}(\hat{Z})\}$ as the feasible set of problem (A.23), and let $Z \in \mathcal{G}_Z^+$ be any reference covariance matrix. Additionally, let $\delta \in (0,1)$ be the desired oracle precision, and define $\varphi(\gamma) = \gamma(\rho^2 + \langle \gamma(\gamma I - \Gamma_Z)^{-1} - I, \hat{Z} \rangle) - \langle Z, \Gamma_Z \rangle$ for any $\gamma > \lambda_{\max}(\Gamma_Z)$. Then, Algorithm A.2 returns in finite time a matrix $L_Z^\delta \in \mathbb{S}_+^d$ with the following properties. (i) Feasibility: $L_Z^\delta \in \mathcal{G}_Z^+$ (ii) $\delta$-Suboptimality: $\langle L_Z^\delta - Z, \Gamma_Z \rangle \geq \delta \max_{L \in \mathcal{G}_Z^+} \langle \Gamma_Z, L - Z \rangle$.*

---

**Algorithm A.2** Bisection algorithm to compute $L_Z^\delta$

---

**Input:** nominal covariance matrix $\hat{Z} \in \mathbb{S}_{++}^d$, radius $\rho \in \mathbb{R}_{++}$,
  reference covariance matrix $Z \in \mathcal{G}_Z^+$,
  gradient matrix $\Gamma_Z \in \mathbb{S}_+^d, \Gamma_Z \neq 0$, precision $\delta \in (0,1)$,
  dual objective function $\phi(\gamma)$ defined in Proposition E.2
1: set $\lambda_1 \leftarrow \lambda_{\max}(\Gamma_Z)$, and let $p_1$ be an eigenvector for $\lambda_1$
2: set $\underline{\gamma} \leftarrow \lambda_1(1 + (p_1^\top \hat{Z} p_1)^{\frac{1}{2}}/\rho)$ and $\overline{\gamma} \leftarrow \lambda_1(1 + \text{Tr}(\hat{Z})^{\frac{1}{2}}/\rho)$
3: **repeat**
4:   set $\tilde{\gamma} \leftarrow (\overline{\gamma} + \underline{\gamma})/2$ and $L \leftarrow (\tilde{\gamma})^2(\tilde{\gamma}I - \Gamma_Z)^{-1}\hat{Z}(\tilde{\gamma}I - \Gamma_Z)^{-1}$
5:   **if** $\frac{d\phi}{d\gamma}(\tilde{\gamma}) < 0$ **then** set $\underline{\gamma} \leftarrow \tilde{\gamma}$ **else**   $\overline{\gamma} \leftarrow \tilde{\gamma}$ **endif**
6: **until** $\frac{d\phi}{d\gamma}(\tilde{\gamma}) > 0$ and $\langle L - Z, \Gamma_Z \rangle \geq \delta\phi(\tilde{\gamma})$
  **Output**: $L$

---

In summary, for any $Z \in \{X_0, W_0, \dots, W_{T-1}, V_0, \dots, V_{T-1}\}$, Algorithm A.2 computes a $\delta$-approximate solutions to the direction-finding subproblem (14) with $\Gamma_Z = \nabla_Z f(W, V)$.

## F. Additional Information on Experiments

**Generation of Nominal Covariance Matrices.** The nominal covariance matrices of the exogenous uncertainties are constructed randomly using the following procedure. For each exogenous uncertainty $z \in \{x_0, w_0, \dots, w_{T-1}, v_0, \dots, v_{T-1}\}$, we denote the dimension of $z$ by $d$ and sample a matrix $M_Z \in \mathbb{R}^{d \times d}$ from the uniform distribution on the hypercube $[0,1]^{d \times d}$. Next, we define $\Xi_Z \in \mathbb{R}^{d \times d}$ as the orthogonal matrix whose columns represent the orthonormal eigenvectors of the symmetric matrix $M_Z + M_Z^\top$. Finally, we set $\hat{Z} = \Xi_Z \Lambda_Z \Xi_Z^\top$, where $\Lambda_Z$ is a diagonal matrix whose main diagonal is sampled uniformly from the interval $[1,2]^d$. The rationale for adopting this cumbersome procedure is to ensure that the covariance matrix $\hat{Z}$ is positive definite.

**Optimality Gap.** The optimality gap of the Frank-Wolfe algorithm visualized in Figure 1b is calculated as the sum of the surrogate optimality gaps $\langle L_Z^\delta - Z, \nabla_Z f(W, V) \rangle$ across all $Z \in \{X_0, W_0 \dots, W_{T-1}, V_0, \dots, V_{T-1}\}$. For more information on the surrogate optimality gaps see [31].

