# OpenReview forum: "Distributionally Robust Linear Quadratic Control"
_NeurIPS.cc/2023/Conference — NeurIPS 2023 spotlight_

### Official Review · Reviewer_iQxV · 2023-07-01

**Soundness:** 3 good
**Presentation:** 3 good
**Contribution:** 3 good
**Rating:** 7
**Confidence:** 3

**Summary:**

The paper **Distributionally Robust Linear Quadratic Control** considers the case of finite-time linear quadratic optimal control with process and measurement noises and known time-varying dynamics. The novelty lies in the fact that the distributions of the initial state and noises are unknown but lie in an ambiguity set defined as a 2-Wasserstein ball centered in a known Gaussian distribution and with known radius. The objective is to minimize the expected LQR cost with the distributions chosen adversarially in that ball. The authors call this adversarial problem "distributionally robust Linear Quadratic Gaussian". It strict generalizes LQG control, which assumes these distributions are known and Gaussian.
The main contribution is twofold. The first part is theoretical, as the authors prove the existence of a Gaussian adversarial distribution, that is, the distributionally robust LQG reduces to classic LQG with unknown Gaussian distributions. A consequence is that the optimal controller is a linear state feedback, like in the classic LQG case. This leads to the second contribution: an algorithm to efficiently estimate the adversarial Gaussian distribution from data. The authors claim that, then, the LQG problem with estimated noise distributions can be solved efficiently by using classical methods.
The authors illustrate convergence of their estimation algorithm on a simulated example.

**Strengths:**

The paper is interesting and the clear exposition makes the reasonings easy to follow. The appendix is well-managed, and I appreciate Appendix A on the solution of classical LQG with a Kalman filter.
I like the idea of allowing for a whole family of noise distributions rather than assuming a fixed one. The main theoretical result helps mitigate the restrictiveness of the commonly-accepted "white noise assumption"; indeed, it shows that allowing for a larger family of noises does not provide any benefits (for distributions close enough to Gaussians). This new problem formulation thus seems relevant. The subsequent algorithm to solve the distributionally robust LQG problem follows naturally and further justifies the interest of the theoretical result. Overall, the reasoning exposed is sound and well-motivated.
Finally, I appreciate that the authors went the extra mile in Section IV by augmenting the theoretical result with a data-efficient algorithm.

**Weaknesses:**

The three main weaknesses of the paper are, in my opinion, 1. the lack of thoroughness of the simulation study; 2. the lack of discussion of the effect of hyperparameters; and 3. some arguments are unclear and should be explicited (although I do not question their conclusions). I detail these three points. These points are not critical for acceptance, but I believe the paper would be improved by addressing them.

W1) The simulation study shows convergence of the algorithm on a randomized use case. This is a good sanity check, but I would also appreciate a simulation of the case when the noise distribution is not normal but still within the Wasserstein ball. The theoretical result ensures that Nature's adversarial strategy _is_ normal, but an illustration of the case when Nature is sub-adversarial would be welcome. In particular, is the cost of the learned policy reduced compared to when the noise is normal?

W2) The role of the choice of ball radii is not discussed. While there is obviously no notion of "optimal" radius, since it simply corresponds to a degree of robustness, I would like to see the evolution in performance with increasing radii. Intuition tells that performance should decrease as robustness increases, but a confirmation in simulation would be welcome.

W3) Some arguments are unclear to me, although their conclusions seem to be valid. In particular:
   1. The authors make multiple claims on convexity of sets of probability measures. Examples are lines 112, 151, 189. I understand that these claims are made by considering Borel measures as a subset of the vector space of signed Borel measures with standard addition and scalar multiplication. I believe this superset should be mentioned explicitly at least ones, as it is rather unusual and notions of convexity of a metric space exist (and differ).
      This is also relevant on line 151, where $\mathcal{W}$ is claimed to be infinite-dimensional despite not being a vector space.
   2. With this understanding of convexity, why is the set $\mathcal{W}$ non convex, as claimed e.g. on lines 112, 151 and 189? As far as I understand, each set $\mathcal{W}_{z}$ is convex, with $z\in\{x_0, w_t, v_t\}$. Then, $\mathcal{W}$ should be convex as the cartesian product of these sets. The only way I understand this non-convexity is if $\mathcal{W}$ is not _equal_ to the cartesian product, but only isometric to this cartesian product by the mapping that computes the marginal distributions. I believe this should be mentioned explicitly, at least in a footnote, as this questions distracted me from more central claims of the paper for a while.
   3. I am unsure about the claim on line 131 that the controller can compute the fictious states $\hat x_0,\dots,\hat x_t$ from the real observations $y_0, \dots, y_t$ without knowing the initial state. Indeed, this would imply in particular that the controller can reconstruct the initial state. Since the time origin is arbitrary, any state could be reconstructed. This claim appear unnecessary for the rest of the argument and should be either removed or clarified in my opinion.
   4. I find the formulation of the first sentence of Proposition 4.1 extremely confusing. In particular, what comes after "then" in the first sentence reads as a logical consequence of what precedes whereas it is actually a definition of the symbols $\mathbb{P}^\star$ and $V_t^\star$. I recommend reformulating.
   5. In Proposition 4.2, I recommend avoiding using the term "smooth". While it has a precise meaning in a specific branch of mathematics, it is often only used informally in control. I would prefer the more standard "infinitely differentiable with $\beta$-Lipschitz gradient".

**Questions:**

My questions are detailed in the above paragraph on weaknesses.  I would appreciate if the authors could respond to these.

**Limitations:**

The authors have adequately addressed limitations and potential negative societal impact of their work.

---

> ### Author Rebuttal · Authors · 2023-08-09
>
> We sincerely appreciate your insightful comments.
> W1) Thank you for suggesting an investigation into the expected cost under different distributions within the ambiguity set. We note that due to the construction of our distributionally robust control problem, the expected cost of our optimal policy under any distribution (Gaussian or not) in the ambiguity set cannot exceed its worst-case expected cost, $i.e.$, the expected cost under the worst-case distribution. However, it is still valuable to investigate how the expected cost evolves under different distributions, so we conducted a numerical experiment along the lines of your suggestion.
> In this experiment, we consider the same setup as in Section 5, where the time horizon is set to $T=2$ and the common radius is set to $\rho = 10$. We generate different distributions that fall within our Wasserstein ambiguity set via a contamination model; specifically, for any $\varepsilon\in[0, 1]$, we compute the $\varepsilon$-contamination distribution ${\mathbb{P}}^{\varepsilon}$ as the Gaussian distribution with mean $0$ and covariance matrix $\Sigma^\varepsilon=\varepsilon\times\Sigma^\star+(1-\varepsilon)\times\hat\Sigma$, where $\hat \Sigma,\Sigma^\star$ denote the covariances matrices of $\hat{\mathbb P}$ and $\mathbb P^\star$, respectively. By leveraging the convexity of the squared Gelbrich distance and the equivalence between the Gelbrich and Wasserstein distances in the case of Gaussian distributions, we can show that $\mathbb{P}^\varepsilon$ belongs to the Wasserstein ambiguity set for all $\varepsilon \in [0, 1]$. We generate Gaussian distributions because checking whether an arbitrary distribution belongs to the Wasserstein ambiguity set is very challenging computationally (in fact, establishing whether an arbitrary discrete distribution falls within ${\mathcal{W}}$ is tantamount to solving a semi-discrete optimal transport problem, which is known to be #P-Hard [R1, Theorem 2.2]).
> * Figure 1b depicts the expected cost of the robust optimal policy $u^*$ under the distribution $\mathbb{P}^\varepsilon$ as a function of $\varepsilon$. Note that the expected cost increases with $\varepsilon$ (as the contaminated distribution approaches the worst-case distribution). Additionally, the expected cost is linear with respect to the contamination level $\varepsilon$, which is aligned with our theoretical results that the objective function of the robust problem admits a linear reformulation in the covariance matrices, as in (9).
> * Figure 1c shows the difference in expected costs between a policy $\hat{u}$ that is optimal under the nominal case (i.e.,that minimizes the expected cost under $\hat{\mathbb P}$) and the robustly optimal policy $u^*$, with both policies evaluated under the contamination distribution $\mathbb{P}^\varepsilon$ for different values of $\varepsilon$. Note that as long as $\varepsilon\geq 0.05$, the robustly optimal policy outperforms the nominal one under $\mathbb{P}^\varepsilon$, therefore resulting in better performance for the vast majority of contamination levels. In addition, even for $\varepsilon\leq 0.05$, the performance of the robust policy is similar to that of the nominal one.
>
> [R1] B. Taskesen, S. Shafieezadeh-Abadeh, and D. Kuhn. "Semi-discrete optimal transport: Hardness, regularization and numerical solution." Math. Prog. 199.1-2 (2023): 1033-1106.
>
> W2) You are correct in noting that the optimal worst-case expected cost is nondecreasing in each of the radii, because increasing any of the radii expands the ambiguity set and relaxes nature's maximization problem. Following your suggestion, we designed a new experiment to quantify the benefits of the robustly optimal policy $u^*$ in comparison to the nominal optimal policy $\hat u$ (that minimizes the expected cost under the nominal distribution) when both are assessed under (i) the nominal distribution and (ii) the corresponding worst-case distributions, respectively, with respect to different radii. We consider the same setup as in Section 5, with a time horizon $T=2$. We vary the common radius $\rho$ from 0 to 10 and we estimate the difference between the expected costs of $\hat{u}$ and the expected costs of $u^*$ under (i) the nominal distribution $\hat{\mathbb P}$ and (ii) under their respective worst-case distributions, which depend on the radius $\rho$. The results, which are shown in Figure 3, illustrate that the gap in expected costs under the worst-case distributions drastically increase as $\rho$ increases, which indicates that the performance of the nominal optimal policy deteriorates rapidly in comparison to the robust policy in worst-case scenarios. In contrast, the gap in expected costs under $\hat{\mathbb P}$ barely changes with the radius $\rho$, which shows that the robust policy has a robust performance in the nominal scenario, almost matching that of the nominal policy (which is optimal under that scenario).
>
> W3)
> * Indeed, the concept of convexity for sets is meaningful only when these sets are part of a linear (vector) space. In our revised manuscript, we will specify that when referring to the convexity of $\mathcal{W}$, where all measures are supported on $\mathbb R^d$, it is implied that $\mathcal{W}$ is a subset of the linear (vector) space of all signed measures on $\mathbb R^d$.
> * In our revised manuscript, we will use $\otimes$ operator instead of $\times$ and make the distinction clear.
> * In the noise-free system (please see lines 127-128), the fictitious initial state is set to $\hat{x}_0 = 0$. In addition, the control input $u_t$ can be computed from observations $y_0,\ldots,y_t$ and does not necessitate knowledge of the true initial state $x_0$. Consequently, the true initial state does not appear in or affect the noise-free system in any manner, and it is not possible to construct the true initial state from this system. We hope our response clarifies your concern.
> * We will clarify the last two points in the revised manuscript.

---

> > ### Comment · Reviewer_iQxV · 2023-08-16
> >
> > Thank you for the response and clarifications. I have no further questions at this stage and will retain my score.

---

### Official Review · Reviewer_BGQp · 2023-07-02

**Soundness:** 3 good
**Presentation:** 3 good
**Contribution:** 3 good
**Rating:** 7
**Confidence:** 4

**Summary:**

The paper proposes a robust method for controlling LQ systems, where process and observation noise distribution laws are unknown, but noise samples are assumed to be independent, zero mean and have their distributions lying close to a a nominal Gaussian distribution in Wasserstein-2 space.

A pair of relaxed settings is used to prove a strong duality between the minimax\maximin problems, showing that the worst-case distribution is Gaussian. A numerical method is proposed for computing this distribution for long horizons.


**Strengths:**

 The paper itself is well-written and gives a thorough overview of the problem.

The fact that the solution for the minimax problem is given by a Gaussian distribution \ linear controller, even if not surprising considering the nature of the quadratic cost and $W_2$/Gelbrich distance, is still not clear from the outset and is important enough from a theoretical point of view. Its proof seems correct as well.


**Weaknesses:**

My main concern here, is that the theoretical contribution is limited to one main result, which is confined to a Gaussian-centered ambiguity set, i.e. the real distribution is assumed to be close to a Gaussian one. This calls for an additional study of other nominal distributions (as the Gelbrich distance equals to $W_2$ for other families of distributions [18], and discussion may be confined to linear filters) or mixtures, or at least for a detailed discussion about the practical implications of this choice (beyond that of the 'concluding remarks' in Sec.6) i.e. about whether this hypothesis may (or might not) be practical in real problems.

However, even if limited, the contribution is still novel and important enough to warrant acceptance.



**Questions:**

Some minor comments \ typos:

It is somewhat misleading to call $\mathcal{W}$ a 'Wasserstein ball', e.g. l.33, while it is not even convex, but I guess that's forgivable since this set is defined clearly.

l. 93 dimension should be $p \times T$.

l.104 it would probably be more didactic to introduce the Wasserstein distance before using it to define  $\mathcal{W}$.


**Limitations:**

The authors addressed the limitations, but a more detailed discussion about the assumption that the nominal distribution is Gaussian is required.

---

> ### Author Rebuttal · Authors · 2023-08-09
>
> * **Extension of Theorem 3.5:** We sincerely appreciate your insightful comment, which has led us to a significant advancement in our work. Specifically, we managed to extend the applicability of our findings to cases where the nominal distribution is an elliptical distribution with finite first- and second-order moments. For more comprehensive information regarding this extension, please refer to the overall response.
>
> * **Wasserstein ball:** Thank you very much for bringing this into our attention. You are correct in asserting that the phrase "Wasserstein ball" could lead to confusion due to the non-convex nature of the ambiguity set $\mathcal W$. The phrase was chosen for its brevity and alignment with customary distributionally robust literature. We will emphasize the non-convex nature of our ambiguity set in the revised paper.
>
> * **Typo:**  Thank you very much for pointing out this typo. We will correct this in the revised manuscript.
>
> * **Definition of Wasserstein distance:** Thank you for bringing this to our attention. We will move the definition of the Wasserstein distance to l.100 in the revised manuscript.

---

> > ### Comment · Reviewer_BGQp · 2023-08-13
> >
> > I thank the authors for their detailed response. I am excited to see you managed to improve your results. Although I believe the new results are sound and extend the theory, it is hard to judge since they haven't been reviewed. I will retain my recommendations.

---

### Official Review · Reviewer_P3Ek · 2023-07-05

**Soundness:** 4 excellent
**Presentation:** 3 good
**Contribution:** 2 fair
**Rating:** 7
**Confidence:** 3

**Summary:**

The paper proposes a distributionally robust version of the output feedback linear quadratic control problem. The goal is to control a system with known partially observed linear dynamics in the face of stochastic disturbances. The stochastic disturbances (both measurement and state disturbances) are drawn from an unknown distribution. This distribution is assumed to belong to some ambiguity set centered at a known gaussian distribution. The disturbances drawn from any distribution in the ambiguity set are assumed to be mean zero, and independent across time, with measurement disturbances independent from the process disturbances. Furthermore, the marginal distribution for the measurement or process disturbance at each time is assumed to be close to the corresponding nominal marginal distribution (as measured by the 2-Wasserstein distance).

Under this setting, the paper finds the optimal output feedback controller for the worst case distribution of disturbances in the ambiguity set. It is found that the worst case distribution of disturbances belonging to the ambiguity set described above is Gaussian. Therefore, the corresponding optimal controller is a Linear-Quadratic Gaussian controller designed for this worst case distribution.

It is then shown that the worst case distribution may be determined in a computationally efficient manner. In particular, it is demonstrated that the Frank-Wolfe algorithm converges to the optimal covariance parameters for the worst case distribution. Furthermore, it is shown that each step of the Frank-Wolfe algorithm may be decomposed into simple, easily parallelizable components.



**Strengths:**

The formulation of the distributionally robust output feedback LQ control problem is novel. It is related to previously published results on distributionally robust state feedback LQ control. As acknowledged by the authors, the output feedback setting brings an extra technical challenge due to the dependence of the optimal state estimator upon the disturbance distribution.

The problem and corresponding solution are clearly presented, and easy to follow.  The authors highlight the rather surprising result that the worst case disturbance distribution from the prescribed ambiguity set is Gaussian.

Given the appropriate context, the results here could be a meaningful step in a unification of worst case and stochastic control.


**Weaknesses:**

The contextualization of the setting studied relative to prior work could be improved. In particular, there are a large class of control synthesis approaches for settings where the noise distributions are either not available or are not Gaussian. In particular, consider H-infinity approaches [Zhou et. al, Robust and Optimal Control, 1996], mixed H-2/H-infinity approaches [(Doyle et. al Optimal Control with Mixed H-2/H-infinity Performance Objectives, 1989) (Bernstein and Haddad,  LQG control with an H-infinity performance bound, 1988)], adversarially robust control [Lee et. al, Performance-Robustness Tradeoffs in Adversarially Robust Control and Estimation, 2023], and nonstochastic control [Hazan and Singh, Introduction to Online Nonstochastic Control, 2023]. It would be useful to discuss several of these results and contrast the setting with the distributionally robust setting.

A thorough discussion about the choice for the ambiguity set and what distributions it can model would be beneficial. In particular, considering mean zero disturbances which are independent across time is restrictive. It fails to model e.g. colored noise.

Despite the fact that it was possible to propose a relatively efficient method to optimize for the worst-case covariance of the noise distribution, the required computation time appears to scale quite poorly with the problem horizon, T. Only short horizons, up to T=20, were considered in the simulations, however many control problems have much longer horizons. This appears to limit the practical applicability.



**Questions:**

Is there any concrete theoretical connection between distributionally robust linear quadratic control and the other methods for incorporating robustness mentioned above (e.g. mixed H-2/H-inf)?

Are there any practical examples where the proposed method substantially outperforms conventional approaches for incorporating robustness to unknown disturbance distributions? Such an example would make the setting more compelling for practical use.

From the experiments, is it possible to detect any clear trends regarding the worst case covariance relative to the nominal covariance of the ambiguity set? E.g. Do we see or expect to see that the worst-case covariances are larger than the central covariances in Loewner order? Combined with the theoretical results, such an observation might justify crude approximate approaches for distributionally robust control design in practice. For example, one could take their estimate for the covariance for the central distribution in the ambiguity set, and simply scale it up by some constant. The resulting covariances could then be used to design a LQG controller.


**Limitations:**

There is no negative societal impact that the authors must address. Limitations to the method that would be worth addressing are mentioned in the weaknesses section. To reiterate, it would be helpful to address:
-	What the ambiguity set can model, and what it cannot
-	In which settings the proposed method would outperform conventional methods from robust control theory for handling unknown disturbance distributions
-	Acknowledging the computational burden of the approach relative to e.g. LQG with a known distribution

---

> ### Author Rebuttal · Authors · 2023-08-09
>
> * **Extension of Theorem 3.5**:  We sincerely appreciate your insightful comment, which has led us to a significant advancement in our work. Specifically, we managed to extend the applicability of our findings to cases where the nominal distribution is an elliptical distribution with finite first- and second-order moments. This extension also involves relaxing the assumption of independence among noise components to uncorrelatedness. Additionally, we can also relax the assumption that the distributions in the ambiguity set have a fixed mean. For more comprehensive information regarding this extension, please refer to the overall response.
> * **Scalability of Algorithm 1**: Thank you very much for pointing us in this direction. Your question helped us realize that our chosen matrix $A$—which was not a convergent matrix ($i.e.$,
> was not satisfying $\lim_{t \to \infty} (A^t)_{ij} = 0$ for all $i, j \in [n]$)—was causing numerical instability in the linearization oracle of Algorithm 1 for $T > 50$. To address this issue, we rescaled the $A$ matrix described in the numerical section by a factor of $0.1$, ensuring it is a convergent matrix. Subsequently, we reran our experiment according to the procedures outlined in Section 5, but this time for $T=100$. The results of this modified experiment are shared in Figure 1a of the attached PDF document.
> * **Relation to the relevant literature**:  Indeed, our work is related to the aforementioned literature, which involves minimizing a worst-case objective functional ($e.g.$, transfer function, cost, regret), where the noise perturbations are selected adversarially [Zhou et. al., Hazan and Singh], or considers mixtures of nominal and worst-case perspectives [Doyle et. al., Bernstein and Hadded, Lee et. al.]. In contrast to these approaches, we adopt a distributionally robust approach, evaluating performance based on the worst-case expected cost in view of all noise distributions close to a nominal one. Although we share the same motivation of enhancing control policies' robustness against uncertainty, we are currently unable to identify any rigorous equivalence or theoretical relationship between our approach and the aforementioned methods. We will investigate further and will incorporate a comprehensive discussion of this related literature in the revised paper.
> * **Loewner order of worst-case covariance matrices**: Assume that we fix the optimal affine controller. Then, the expected cost can be reformulated as $\mathbb E_{\mathbb{P}}[\xi^\top S\xi]$ for some matrix $S \succeq 0$, where $\xi\sim {\mathbb{P}}$ encapsulates the uncertainties inherent to the model with  $\mathbb E_{\mathbb{P}}[\xi]=0$ and $\mathbb E_{\mathbb{P}}[\xi \xi^\top]=\Sigma$. Let $\hat\Sigma$ denote the nominal covariance matrix. Then, by Theorem 3.5 maximizing the expected cost over a 2-Wasserstein ambiguity set $\mathcal{W}$ of radius $\rho$ is equivalent to maximizing $Tr(S \Sigma)$ over all $\Sigma\succeq 0$ satisfying $G(\Sigma, \hat \Sigma)\leq\rho^2$. In the following, we will show that the worst-case covariance matrix cannot be smaller than the nominal covariance matrix $\hat\Sigma$ with respect to the Loewner order. By applying a linear coordinate transformation, we may assume without loss of generality that $\hat\Sigma=I$. Consider now any matrix $\Sigma\succeq 0$ in the Gelbrich ball, that is, $G(\Sigma, I)^2 = Tr(I+\Sigma-2\Sigma^{1/2})\leq \rho^2$. Using the spectral decomposition $\Sigma=\sum_{i=1}^n\lambda_i v_i v_i^\top$, where $\lambda_i\geq 0$ is the $i$-th eigenvalue and $v_i$ is the corresponding eigenvector (we may assume the eigenvectors are orthonormal), the squared Gelbrich distance can be reformulated as $Tr(I+\Sigma-2\Sigma^{1/2}) = \sum_{i=1}^n (1-\lambda_i^{1/2})^2$. Note that $\hat \Sigma = I$ has eigenvalues 1. In the following, we will show that if $\Sigma$ has any eigenvalues strictly less than 1, it will not attain the maximum value of $Tr(\Sigma S)$, that is, there exists $\Sigma'$ satisfying $G(\Sigma', I) \leq \rho$ with eigenvalues greater or equal than 1. Now let $\Sigma'=\sum_{i=1}^n \max${$1,\lambda_i$}$v_i v_i^\top$, which has the same eigenvectors as $\Sigma$, but all eigenvalues smaller than 1 are set to 1, that is, $\Sigma'$ has larger eigenvalues than $\Sigma$ with the same eigenvectors. The matrix $\Sigma'$ is feasible because $Tr(I+\Sigma'-2(\Sigma')^{1/2}) = \sum_{i=1}^n (1-\max${$1,\lambda_i$}$^{1/2})^2 \leq \sum_{i=1}^n (1-\lambda_i^{1/2})^2 = Tr(I+\Sigma-2\Sigma^{1/2}) \leq \rho^2$. In addition, the expected loss with respect to $\Sigma'$ is at least as large as the expected loss with respect to $\Sigma$. Formally, we have $Tr(S\Sigma') \geq Tr(S\Sigma)$ because $S\succeq 0$ and because $\Sigma'\succeq \Sigma$ by construction. This shows that the worst-case covariance matrix must be non-inferior to the nominal covariance matrix in Loewner order. Similar arguments can be used to show that increasing $\rho$, increases the worst-case covariance matrix in Loewner order. In Figures 2a and 2b shared in the uploaded PDF file, for $n=m=p=2$, we illustrate the worst-case covariance matrices for 10 different values of $\rho$ split evenly in the range $[0,1]$ satisfying $\rho_{x_0} = \rho_{w_0} = \ldots= \rho_{w_{T-1}} = \rho_{v_0}=\ldots=\rho_{v_{T-1}}=\rho$.  We can see empirically from the plots that indeed the worst-case covariance matrices are inflating as the radius $\rho$ of the balls increases. Interestingly, Figure 2 indicates that scaling the empirical covariance matrix $\hat X_0$ might approximate its worst-case in this experiment, but the same scaling wouldn't provide an accurate estimate for the worst-case covariance of $v_0$.

---

> > ### Comment · Reviewer_P3Ek · 2023-08-16
> >
> > Thank you - your response mostly addresses my questions.  While I will raise my score, I do want to see better physical justification for the chosen ambiguity set.

---

> > > ### Author Response · Authors · 2023-08-17
> > >
> > > Thank you very much for raising your score. We will provide additional justification for using Wasserstein ambiguity sets in the final version of the paper.

---

### Official Review · Reviewer_RqGY · 2023-07-08

**Soundness:** 3 good
**Presentation:** 3 good
**Contribution:** 3 good
**Rating:** 7
**Confidence:** 4

**Summary:**

- The paper considers a standard LQ setup with uncertainty in the distributions of system noise, observation noise and initial state.
- Described as Zero-Sum Game between the controller and nature, they show the optimal decisions for both players. Specifically they show that the worst-case distribution is Gaussian and the optimal control law is linear.
- They provide an numerically efficient approach to solve the distributionally robust LQ control problem based on Frank-Wolfe algorithm.
-  Simulation experiments are provided to show the computational efficiency of the proposed algorithm compared to MOSEK.

**Strengths:**

The results provided in the manuscript make multiple important contribution.

- The optimal control law still remains linear and the worst case distribution is still Gaussian.
- The proof technique is novel which relies on the "purified states" instead of usual dynamic programming approaches, it is interesting to see how this approach can be used in other LQG problems.
- Simulation results show the computational efficiency of the Frank-Wolfe algorithm over MOSEK.


**Weaknesses:**

- There are no obvious major weakness in the manuscript. Some potential minor weakness are currently mentioned in the form of questions in the comment below.


**Questions:**

- In general the adaptive linear quadratic control results are provided under assumption of sub-gaussian system noises (Assumption A1, http://proceedings.mlr.press/v19/abbasi-yadkori11a/abbasi-yadkori11a.pdf). Is it possible to extend these results to Sub-Gaussian nominal distribution?

**Limitations:**

Authors provide some limitations in form of possible extensions and future work in the last section.

Social Impact: NA

---

> ### Author Rebuttal · Authors · 2023-08-09
>
> We sincerely appreciate your insightful comment, which has led us to a significant advancement in our work. Specifically, we managed to extend the applicability of our findings to cases where the nominal distribution is an elliptical distribution with finite first- and second-order moments. This extension also involves relaxing the assumption of independence among noise components to uncorrelatedness. Therefore, our results extend to the sub-Gaussian distributions that are elliptical. For more comprehensive information regarding this extension, please refer to the overall response.

---

> > ### Comment · Reviewer_RqGY · 2023-08-10
> > **Response to the Rebuttal**
> >
> > I thank the authors for their responses. After reading their rebuttal, I will retain my recommendation regarding this paper.

---

### Author Rebuttal · Authors · 2023-08-09

We express our gratitude to the reviewers for their thoughtful comments and questions.

In the reviews, a common concern was raised about our main result in Theorem 3.5. This theorem establishes the optimality of a linear control policy and identifies the worst-case distribution as a Gaussian distribution. However, our proof was specifically formulated under the assumption of a Gaussian nominal distribution.

Thanks to the insightful feedback provided by the reviewers, we have identified the potential for extending Theorem 3.5. Specifically, we can **extend Theorem 3.5** to situations where **the nominal distribution is an elliptical distribution with finite first- and second-order moments** and show that a linear control policy is optimal while the worst-case distribution is an elliptical distribution. In addition, we can **relax the independence assumption** of the noise components **to uncorrelatedness** in this case.
Finally, we can also **drop the zero mean assumption for the nominal distribution** and **relax the assumption that the distributions in the ambiguity set have a fixed mean**.


The class of elliptical distributions generalizes multivariate Gaussian distributions and spherical distributions [R1] and includes symmetric distributions with light and heavy tails.
Examples of elliptical distributions include the Laplace, logistic, and $t$-distribution, among others.
Therefore, even though we cannot extend our results readily to any sub-Gaussian distribution as asked by Reviewer RqGy, we can extend our results to some sub-Gaussian distributions that are also elliptical.

There are two key reasons that allow us to extend Theorem 3.5 to situations where the nominal distribution is elliptical with finite first- and second-order moments. Firstly, when ${\mathbb{P}}$ is an elliptical distribution, the conditional expectation $\mathbb E_{\mathbb{P}}[\xi | \xi' = H\xi ]$ is linear in $\xi'$ [R2, Theorem 4]. This implies that under any fixed elliptical distribution, a linear control policy is optimal. Consequently, this allows us to establish a parallel lower-bound problem to (10), wherein $\mathcal W_{\mathcal N}$ is substituted by a subset of the ambiguity set $\mathcal W$ containing only elliptical distributions sharing the same characteristic function with the nominal one. This observation permits us to focus on linear policies within the inner minimization of the lower-bound problem without loss of generality.
Secondly, the distance between two elliptical distributions, sharing identical characteristic functions, is equivalent to the Gelbrich distance between their corresponding mean vectors and covariance matrices [R3, Theorem 2.4]. This observation further empowers us to substitute the feasible set of the outer maximization with the Gelbrich ambiguity set. The remainder of our proof seamlessly adapts to elliptical distributions, as other arguments do not rely on the Gaussian structure of the nominal distribution.

Furthermore, we can also extend our numerically efficient method to compute the worst-case distribution, now elliptical, along with the optimal linear control policy, to situations where the nominal distribution is elliptical. Notably, Proposition 4.1's applicability naturally extends the case where the worst-case distribution ${\mathbb{P}}^\star$ becomes an elliptical distribution sharing the same characteristic function with the nominal one. By leveraging our Frank-Wolfe algorithm, we can compute the covariance matrix of this distribution. Subsequently, solving a linear quadratic control problem using ${\mathbb{P}}^\star$ allows us to determine the optimal controller. Remarkably, even in this context where the nominal distribution is elliptical, the separation principle remains applicable [R4], and the recursive equations of the Kalman filter continue to hold for elliptical distributions [R5, R6].

The departure from the fixed mean assumption in the ambiguity set can be inferred from [R7, Theorem 2.7], [R8, Theorem 2.16], and [R7, Theorem 3.5]. However, to maintain focus on the core message of the paper and prevent undue complexity in terms of calculations and notation, we did not include this extension in the submitted version of our manuscript.

We thank the reviewers for recommending us to explore this meaningful direction that strengthens the main result of our paper.

[R1] R.D. Lord, The Use of the Hankel Transform in Statistics I. General Theory and Examples, Biometrika, vol. 41, no. 1/2, 1954, pp. 44–55, JSTOR.


[R2] K.-C. Chu, Estimation and decision for linear systems with elliptical random processes, IEEE Conference on Decision and Control on Adaptive Processes, 1972, pp. 647-651.


[R3] M. Gelbrich, On a formula for the $L^2$-Wasserstein metric between measures on Euclidean and Hilbert spaces, Mathematische Nachrichten, 147 (1990), pp. 185–203.



[R4] Witsenhausen, Hans S., Separation of estimation and control for discrete time systems, Proceedings of the IEEE 59.11 (1971): 1557-1566.

[R5] Basu, A. K., \& Das, J. K. (1994). A Bayesian Approach to Kalman Filter for Elliptically Contoured Distribution and its Application in Time Series Models. Calcutta Statistical Association Bulletin, 44(1–2), 11–28.

[R6] Girón, F. J., \& Rojano, J. C. (1994). Bayesian Kalman Filtering with Elliptically Contoured Errors. Biometrika, 81(2), 390–395.


[R7] V. A. Nguyen, S. Shafieezadeh-Abadeh, D. Kuhn, \& P. Mohajerin Esfahani, Bridging Bayesian and minimax mean square error estimation via Wasserstein distributionally robust optimization. Mathematics of Operations Research, 2023, 48(1), 1-37.

[R8] K.-T. Fang, S. Kotz, and K. W. Ng, Symmetric Multivariate and Related Distributions, Chapman \& Hall, 1990.

---

### Decision · Program_Chairs · 2023-09-21

**Decision:**

Accept (spotlight)

**Comment:**

**Summary:** This paper consider a generalization of the discrete-time, finite-horizon LQG problem, where the noise distributions are unknown and belong to Wasserstein ambiguity sets centered at nominal (Gaussian) distributions. The goal is to minimize a worst-case cost across all distributions in the ambiguity set, including non-Gaussian distributions. The main contribution of this paper are in showing that the optimal control policy is linear (w.r.t. observations), and in providing a numerical approach to compute this optimal policy.

**Comments:** The review scores are 7, 7, 7, 7. Reviewers are unanimously positive about the technical contributions of this paper. This paper will be of interest to the NeurIPS community, especially for the subareas of control+learning / reinforcement learning. For the final version,  please update your paper based on the comments and the suggestions by the reviewers.

Thanks!

AC